# EFFICIENT LOCAL LINEARITY REGULARIZATION TO OVERCOME CATASTROPHIC OVERFITTING

**Elias Abad Rocamora**[1,*], **Fanghui Liu**[2,†], **Grigorios G. Chrysos**[3,†],
**Pablo M. Olmos**[4], **Volkan Cevher**[1]
[1]LIONS - École Polytechnique Fédérale de Lausanne, [2]University of Warwick,
[3]University of Wisconsin-Madison, [4]Universidad Carlos III de Madrid

## ABSTRACT

Catastrophic overfitting (CO) in single-step adversarial training (AT) results in abrupt drops in the adversarial test accuracy (even down to $0\%$). For models trained with multi-step AT, it has been observed that the loss function behaves locally linearly with respect to the input, this is however lost in single-step AT. To address CO in single-step AT, several methods have been proposed to enforce local linearity of the loss via regularization. However, these regularization terms considerably slow down training due to *Double Backpropagation*. Instead, in this work, we introduce a regularization term, called `ELLE`, to mitigate CO *effectively* and *efficiently* in classical AT evaluations, as well as some more difficult regimes, e.g., large adversarial perturbations and long training schedules. Our regularization term can be theoretically linked to curvature of the loss function and is computationally cheaper than previous methods by avoiding *Double Backpropagation*. Our thorough experimental validation demonstrates that our work does not suffer from CO, even in challenging settings where previous works suffer from it. We also notice that adapting our regularization parameter during training (`ELLE-A`) greatly improves the performance, specially in large $\epsilon$ setups. Our implementation is available in `https://github.com/LIONS-EPFL/ELLE`.

## 1 INTRODUCTION

Adversarial Training (AT) (Madry et al., 2018) and TRADES (Zhang et al., 2019) have emerged as prominent training methods for training robust architectures. However, these training mechanisms involve solving an inner optimization problem per training step, often requiring an order of magnitude more time per iteration in comparison to standard training (Xu et al., 2023). To address the computational overhead per iteration, the solution of the inner maximization problem in a single step is commonly utilized. While this approach offers efficiency gains, it is also known to be unstable (Tramèr et al., 2018; Shafahi et al., 2019; Wong et al., 2020; de Jorge et al., 2022).

Indeed, the single-step AT approach results in the so-called *Catastrophic Overfitting (CO)* as the adversarial perturbation size $\epsilon$ increases (Wong et al., 2020; Andriushchenko and Flammarion, 2020). CO is characterized by a sharp decline (even down to $0\%$) in multi-step test adversarial accuracy and a corresponding spike (up to $100\%$) in single-step train adversarial accuracy.

An important property of adversarially robust models is local linearity of the loss with respect to the input (Ross and Doshi-Velez, 2018; Simon-Gabriel et al., 2019; Moosavi-Dezfooli et al., 2019; Qin et al., 2019; Andriushchenko and Flammarion, 2020; Singla et al., 2021; Srinivas et al., 2022; Li and Spratling, 2023). Explicitly enforcing local linearity has been shown to allow reducing the number of steps needed to solve the inner maximization problem, while avoiding CO and gradient obfuscation (Qin et al., 2019; Andriushchenko and Flammarion, 2020). Nevertheless, all existing methods incur a $\times 3$ runtime due to *Double Backpropagation* (Etmann, 2019) Given this time-consuming operation to avoid CO, a natural question arises:

*Can we efficiently overcome catastrophic overfitting when enforcing local linearity of the loss?*

---

[*]Partially done at Universidad Carlos III de Madrid, correspondance: `elias.abadrocamora@epfl.ch`
[†]Partially done at LIONS-EPFL

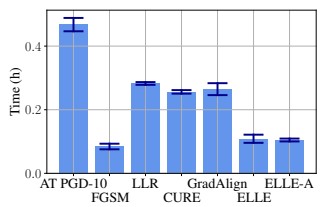

| $\epsilon$ | | 8 | | 16 | |
|---|---|---|---|---|---|
| Method | AA | Clean | AA | Clean |
| LLR | $42.18 \pm (0.20)$ | $75.02 \pm (0.09)$ | $16.92 \pm (0.20)$ | $42.81 \pm (9.62)$ |
| CURE | $43.60 \pm (0.17)$ | $77.74 \pm (0.11)$ | $\underline{18.25} \pm (0.45)$ | $52.49 \pm (0.04)$ |
| GradAlign | $\mathbf{44.66} \pm (0.21)$ | $\mathbf{80.50} \pm (0.07)$ | $17.46 \pm (1.71)$ | $44.35 \pm (15.32)$ |
| ELLE | $42.78 \pm (0.95)$ | $\underline{80.13} \pm (0.32)$ | $\mathbf{18.28} \pm (0.17)$ | $\mathbf{59.73} \pm (0.16)$ |
| ELLE-A | $\underline{44.32} \pm (0.04)$ | $79.81 \pm (0.10)$ | $18.03 \pm (0.15)$ | $\underline{59.21} \pm (1.23)$ |
| AT PGD-10 | $46.95 \pm (0.11)$ | $79.11 \pm (0.08)$ | $24.77 \pm (0.26)$ | $59.64 \pm (0.46)$ |

(a) Runtime comparison                          (b) CIFAR10

Figure 1: Comparison against single-step methods enforcing local linearity. We train with our method `ELLE` and its adaptive regularization variant `ELLE-A`. The multi-step AT PGD-10 results are included for comparison. We measure **(a)** the average total runtime and FGSM training as the fast but $0\%$ AA accuracy baseline. **(b)** The clean and AutoAttack accuracies. We mark the best method and the runner-up in **bold** and underlined respectively. Our methods, `ELLE` and `ELLE-A`, attain the best or comparable AA accuracy while employing less than $50\%$ of the time of previous methods.

In this work, we answer this question affirmatively. In particular, we propose *Efficient Local Linearity Enforcement* (`ELLE`) , a plug-in regularization term that encourages local linearity, which is able to obtain state-of-the-art adversarial accuracy while avoiding CO. Our algorithm is based on a key property of locally linear functions. Let $\boldsymbol{x}_a, \boldsymbol{x}_b \in \mathcal{X} \subseteq \mathbb{R}^d$ with $\mathcal{X}$ a convex set, the function $\boldsymbol{h} : \mathbb{R}^d \rightarrow \mathbb{R}$ is *locally linear in $\mathcal{X}$* if and only if:

$$\boldsymbol{h}((1 - \alpha) \cdot \boldsymbol{x}_a + \alpha \cdot \boldsymbol{x}_b) = (1 - \alpha) \cdot \boldsymbol{h}(\boldsymbol{x}_a) + \alpha \cdot \boldsymbol{h}(\boldsymbol{x}_b) \ \forall \alpha \in [0, 1], \ \forall \boldsymbol{x}_a, \boldsymbol{x}_b \in \mathcal{X} . \quad (1)$$

Our method is based on enforcing Eq. (1) for the loss function $\mathcal{L}$, uniformly in the convex set $\{\boldsymbol{x} : ||\boldsymbol{x} - \boldsymbol{x}_i||_\infty \leq \epsilon\}$ around training points $\boldsymbol{x}_i$.

Our main contribution is the proposition of a novel single-step AT training paradigm `ELLE` based on the principle of the local linearity of the loss. Our regularization term does not require differentiating gradients and avoids *Double Backpropagation*, representing a much more efficient alternative to the existing methods, see Fig. 1a. Besides, we theoretically analyze the relationship between the local, linear approximation error in `ELLE` and the curvature of the loss. This allows detecting the appearance of CO when the local, linear approximation error suddenly increases and avoid CO when regularizing this measure. We evaluate `ELLE` on popular benchmarks CIFAR10/100, SVHN and ImageNet. In particular, for the PreActResNet18 architecture, $\epsilon = 12/255$ when adding our regularization term, N-FGSM increases its AutoAttack accuracy from $21.18\%$ to $26.35\%$, see Table 2.

A side-benefit of our method is that `ELLE` can overcome CO for short and long training schedules (see Sec. 4.1) and small and large $\epsilon$, whereas other single-step AT methods suffer from CO for long schedules and/or large $\epsilon$. We denote this phenomenon as *Delayed CO*, see Sec. 4.1. Moreover, we find that adapting our regularization parameter to the local linearity of the network during training (`ELLE-A`) and combining with other methods as GAT (Sriramanan et al., 2020) or N-FGSM (de Jorge et al., 2022) helps avoiding CO and improves their performance, specially for large $\epsilon$, see Sec. 4.5.

**Notation:** We use the shorthand $[n] := \{1, 2, \ldots, n\}$ for a positive integer $n$. We use bold capital (lowercase) letters, e.g., $\boldsymbol{X}$ ($\boldsymbol{x}$) for representing matrices (vectors). The $j^{\text{th}}$ column of a matrix $\boldsymbol{X}$ is given by $\boldsymbol{x}_{:j}$. The element in the $i^{\text{th}}$ row and $j^{\text{th}}$ column is given by $x_{ij}$, similarly, the $i^{\text{th}}$ element of a vector $\boldsymbol{x}$ is given by $x_i$. The $\ell_\infty$, and $\ell_p$ norms of a vector $\boldsymbol{x} \in \mathbb{R}^d$ for $1 \leq p < \infty$ are given by: $||\boldsymbol{x}||_\infty = \max_{i \in [d]} |x_i|$ and $||\boldsymbol{x}||_p = \left( \sum_{i=1}^d |x_i|^p \right)^{1/p}$ respectively. Lastly, we denote the closed ball centered at $\boldsymbol{x}$, of radius $\epsilon$ in the $\ell_p$ norm as $\mathbb{B}[\boldsymbol{x}, \epsilon, p] = \{\boldsymbol{z} : ||\boldsymbol{x} - \boldsymbol{z}||_p \leq \epsilon\}$.

## 2 BACKGROUND AND RELATED WORK

In Sec. 2.1 we summarize the challenges of making Robust Training computationally efficient and in Sec. 2.2, we cover the efforts towards addressing them. We assume our dataset $\{(\boldsymbol{x}_i, y_i)\}_{i=1}^N$ sampled from a unknown distribution $\mathcal{D}$ in the space $[0, 1]^d \times [o]$, where $o$ is the number of classes.

## 2.1 COMPUTATIONALLY EFFICIENT ROBUST TRAINING

In order to make AT more computationally efficient, several approaches have been proposed. In Free AT (Shafahi et al., 2019), the network parameters $\theta$ and adversarial perturbation $\delta$ are simultaneously updated for several steps within the same batch. In Fast AT (Wong et al., 2020), a single step FGSM attack (Goodfellow et al., 2015) is performed at a randomly sampled point $x + \eta$ with $\eta \sim \mathrm{Unif}(-\epsilon, \epsilon)$. These methods alleviate the cost of solving the inner maximization problem and considerably speed up training.

## 2.2 CATASTROPHIC OVERFITTING (CO)

The speed up in single-step AT emerges at a cost. As initially found by Wong et al. (2020) for standard FGSM training, and later reported by Andriushchenko and Flammarion (2020) in Free AT and Fast AT. Adversarial test accuracy dramatically drops (even to 0 sometimes) within a few epochs. This phenomenon is known as *Catastrophic Overfitting (CO)*. Several efforts have been made towards understanding CO (Li et al., 2020; Andriushchenko and Flammarion, 2020; Kim et al., 2021; Ortiz-Jimenez et al., 2023; He et al., 2023), with a sudden change in the local linearity of the loss with respect to the input as the most common explanation. In the following, we summarize the methods aiming at avoiding CO:

**Regularization based.** Andriushchenko and Flammarion (2020) propose GradAlign: penalizing gradient missalignment to avoid CO. Qin et al. (2019) propose LLR: regularizing the worst case of the first order Taylor approximation along any direction $||\delta||_\infty \leq \epsilon$. However, the computation of the gradient of the regularization terms in GradAlign and LLR with respect to the model parameters $\theta$ involves *Double Backpropagation* (Etmann, 2019), which leads to a $3\times$ per-epoch training time in comparison to FGSM training. More recently, Sriramanan et al. (2020) propose GAT, a new regularization term and attack based on penalizing the difference between logits at the training point and a random sample. Sriramanan et al. (2021) introduce a regularization term penalizing the nuclear norm of the logits within the batch. Nevertheless, GAT and NuAT still suffer from CO under a larger $\epsilon$ (de Jorge et al., 2022).

**Noise based.** Tramèr et al. (2018) add random noise prior to the gradient computation in FGSM training. Wong et al. (2020) revisit FGSM training and show that adding random uniform noise to the input before the FGSM attack makes single step AT plausible (Fast AT). Nevertheless, it was later shown that for larger adversarial perturbations ($\epsilon$), Fast AT suffers from CO (Andriushchenko and Flammarion, 2020). In the same direction, de Jorge et al. (2022) find doing data augmentation by adding random noise $||\eta||_\infty \leq 2 \cdot \epsilon$ makes FGSM training feasible. We find N-FGSM suffers from CO for large $\epsilon$ and/or long training schedules, see Figs. 3 and 4.

**Other.** Park and Lee (2021) perform adversarial attacks in the latent space. Tsiligkaridis and Roberts (2022) propose using Frank-Wolfe optimization for AT and link the distortion of the loss landscape with the $\ell_2$ norm of the adversarial perturbations. Li et al. (2022) tackle CO by constraining the network weights to a subspace given by the principal components of AT trained weights. Zhang et al. (2022b) reformulate the AT problem as a Bilevel Optimization problem and propose a single-step approach to solve it. Golgooni et al. (2023) propose setting to zero the adversarial perturbation coordinates that fall bellow a threshold, however de Jorge et al. (2022) show this leads to a performance degradation for large $\epsilon$.

## 3 METHOD

In this section, we firstly introduce our regularization term and training algorithm, `ELLE(-A)`, which efficiently enforces local linearity of the loss function to avoid CO. Sequentially, we elucidate the benefits of our algorithm when compared to previous methods.

### 3.1 ALGORITHM DESCRIPTION

Motivated by Eq. (1), our regularization term is designed to enforce local linearity, which can be described by the local, linear approximation error:

---

**Algorithm 1** `ELLE` (`ELLE-A`) adversarial training. Pseudo-code in teal is only run for `ELLE-A`. $\mu$ and $\sigma$ represent the mean and standard deviation respectively.

1: **Inputs:** # epochs $T$, # batches $M$, radius $\epsilon$, regularization parameter $\lambda$ and decay rate $\gamma$.
2: err_list $= []$, $\lambda_{\max} = \lambda$, $\lambda = 0$
3: **for** $t = 1, \ldots, T$ **do**
4:   **for** $i = 1, \ldots, M$ **do**
5:    $\boldsymbol{x}_{\text{FGSM}}^i = \boldsymbol{x}^i + \epsilon \cdot \text{sign}\left(\nabla_{\boldsymbol{x}^i} \mathcal{L}(\boldsymbol{f_\theta}(\boldsymbol{x}^i), y^i)\right)$     ▷ Standard FGSM attack
6:    $\boldsymbol{x}_a^i, \ \boldsymbol{x}_b^i \sim \boldsymbol{x}^i + \text{Unif}\left([-\epsilon, \epsilon]^d\right)$       ▷ Random samples
7:    $\alpha \sim \text{Unif}\left([0, 1]\right)$
8:    $\boldsymbol{x}_c^i = (1 - \alpha) \cdot \boldsymbol{x}_a^i + \alpha \cdot \boldsymbol{x}_b^i$    ▷ Linear (convex) combination of $\boldsymbol{x}_a^i$ and $\boldsymbol{x}_b^i$
9:    $E_{\text{lin}} = \left| \mathcal{L}(\boldsymbol{f_\theta}(\boldsymbol{x}_c^i), y^i) - (1 - \alpha) \cdot \mathcal{L}(\boldsymbol{f_\theta}(\boldsymbol{x}_a^i), y^i) - \alpha \cdot \mathcal{L}(\boldsymbol{f_\theta}(\boldsymbol{x}_b^i), y^i) \right|^2$
10:    **if** $E_{\text{lin}} > \mu(\text{err\_list}) + 2 \cdot \sigma(\text{err\_list})$ **then** $\lambda = \lambda_{\max}$
11:    **else** $\lambda = \gamma \cdot \lambda$
12:    err_list $=$ err_list $+ [E_{\text{lin}}]$
13:    $\nabla_{\boldsymbol{\theta}} = \nabla_{\boldsymbol{\theta}} \mathcal{L}(\boldsymbol{f_\theta}(\boldsymbol{x}_{\text{FGSM}}^i), y^i) + \lambda \nabla_{\boldsymbol{\theta}} E_{\text{lin}}$     ▷ Compute parameter gradients
14:    $\boldsymbol{\theta} = \text{optimizer}(\boldsymbol{\theta}, \nabla_{\boldsymbol{\theta}})$     ▷ Standard parameters update, (e.g. SGD)

---

**Definition 1** (local, linear approximation error). *Let $\boldsymbol{h} : \mathbb{R}^d \to \mathbb{R}^o$, the local, linear approximation error of $\boldsymbol{h}$ at $\mathbb{B}[\boldsymbol{x}, \epsilon, p]$ for $1 \leq p \leq \infty$ is given by:*

$$E_{Lin}(\boldsymbol{h}, \boldsymbol{x}, p, \epsilon) = \mathop{\mathbb{E}}_{\substack{\boldsymbol{x}_a, \boldsymbol{x}_b \sim Unif(\mathbb{B}[\boldsymbol{x}, \epsilon, p]) \\ \alpha \sim Unif([0,1])}} \left[ \|\boldsymbol{h}((1 - \alpha) \cdot \boldsymbol{x}_a + \alpha \cdot \boldsymbol{x}_b) - (1 - \alpha) \cdot \boldsymbol{h}(\boldsymbol{x}_a) - \alpha \cdot \boldsymbol{h}(\boldsymbol{x}_b)\|_2 \right]. \quad (2)$$

Our goal is to minimize Eq. (2). Our algorithm, `ELLE`, and its adaptive regularization variant, `ELLE-A`, are described in Algorithm 1. `ELLE(-A)` combine a single step FGSM attack with a regularization term consisting on a single $(\boldsymbol{x}_a, \boldsymbol{x}_b, \alpha)$-sample for Eq. (2) with $\boldsymbol{h}$ as the Cross-Entropy loss, see line 9 in Algorithm 1. Furthermore, the geometric intuition behind our proposed local, linear approximation error is presented below.

**Adaptive local linerarity regularization** (`ELLE-A`): In the initial 15 epochs in Fig. 2, the network trained with `ELLE` reports a $\times 10$ smaller $E_{\text{lin}}$ than the one trained with FGSM. Nevertheless, in these early stages FGSM training has not suffered from CO yet. This suggests regularization is not needed all the time across epochs. In order to use a less invasive regularization scheme, we propose to adapt $\lambda$ across steps. `ELLE-A` adapts the value of the regularization parameter $\lambda$ in order to perform a less invasive regularization, see teal color in Algorithm 1. Since CO appears when the loss suddenly becomes non-locally-linear (see Fig. 2 and Andriushchenko and Flammarion (2020)), we propose initializing $\lambda = 0$ and increasing to $\lambda = \lambda_{\max}$ every time an unusual value of $E_{\text{lin}}$ is encountered, this adaptive strategy is depicted in teal in Algorithm 1.

### 3.2 THEORETICAL UNDERSTANDING

The local, linear approximation error can be linked with second order directional derivatives when evaluating smooth-enough functions. To this end, we define the second order directional derivative.

**Definition 2** (Second order directional derivative $D_{\boldsymbol{v}}^2(h_i(\boldsymbol{x}))$). *Let $\boldsymbol{h} \in C^2(\mathbb{R}^d)$ be a twice differentiable mapping, the second order directional derivative of the $i^{th}$ output $h_i$ along direction $\boldsymbol{v}$ at input $\boldsymbol{x}$ is given by:*

$$D_{\boldsymbol{v}}^2(h_i(\boldsymbol{x})) = \boldsymbol{v}^\top \nabla_{\boldsymbol{xx}}^2 h_i(\boldsymbol{x}) \boldsymbol{v}. \quad (3)$$

In the case of three times differentiable classifiers, we have the following connection of our regularization term with the second order directional derivatives, with the proof deferred to Appx. C.

**Proposition 1.** *Let $\boldsymbol{h} \in C^3(\mathbb{R}^d)$ be a three times differentiable mapping. Let $E_{Lin}(\boldsymbol{h}, \boldsymbol{x}, p, \epsilon)$ and $D_{\boldsymbol{v}}^2(h_i(\boldsymbol{x}))$ be defined as in Definitions 1 and 2 and $\left[D_{\boldsymbol{v}}^2(h_i(\boldsymbol{x}))\right]_{i=1}^o \in \mathbb{R}^o$ be the vector containing*

*the second order directional derivatives along direction $\boldsymbol{v}$ for every output coordinate of $\boldsymbol{h}$. Then, the following relationship follows:*

$$E_{Lin}(\boldsymbol{h}, \boldsymbol{x}, p, \epsilon) = \mathop{\mathbb{E}}_{\substack{\boldsymbol{x}_a, \boldsymbol{x}_b \sim Unif(\mathbb{B}[\boldsymbol{x}, \epsilon, p]) \\ \alpha \sim Unif([0,1])}} \left( \left\| \left[ \frac{-\alpha(1-\alpha)}{2} D^2_{\boldsymbol{x}_a - \boldsymbol{x}_b}(h_i(\boldsymbol{x}_c)) + O(||\boldsymbol{x}_a - \boldsymbol{x}_b||_\infty^3) \right]_{i=1}^o \right\|_2 \right) , \quad (4)$$

*where $\boldsymbol{x}_c := (1-\alpha) \cdot \boldsymbol{x}_a + \alpha \cdot \boldsymbol{x}_b$.*

**Remark 1.** *More accurate approximations of $D^2_{\boldsymbol{x}_a - \boldsymbol{x}_b}$ and their corresponding local, linear approximation error definitions can be obtained by utilizing more than $3$ points. Nevertheless, in our experiments we observe that $3$ points are enough to overcome CO. An analysis with more than $3$ points is available in Appx. C.*

Proposition 1 builds a connection between our regularization term / local, linear approximation error and second order directional derivatives, i.e., curvature. Even though curvature is not defined for popular classifiers living in $C^0$ such as ReLU NNs, other popular architectures such as Transformers or NNs with smooth activations living in $C^\infty$ indeed have curvature and our method will be implicitly enforcing low-curvature for them. Moreover, for $C^\infty$ classifiers, it might not be computationally feasible to compute $D^2_{\boldsymbol{x}_a - \boldsymbol{x}_b}(h_i(\boldsymbol{x}_c))$ whereas our regularization term is efficiently computed. In Sec. 3.3 we compare our method with other methods directly enforcing local linearity. We perform a similar analysis with the LLR regularization term (Qin et al., 2019) in Appx. C, however LLR is a much more expensive regularization term as shown in Fig. 1. We further analyze other possible regularization terms in Appendices B.11 and B.14.

### 3.3 COMPARISON WITH EXPLICIT LOCAL-LINEARITY ENFORCING ALGORITHMS

Relating local linearity with robustness, Simon-Gabriel et al. (2019); Ross and Doshi-Velez (2018) propose penalizing the gradient norm at the training sample, i.e., introducing the regularization term $||\nabla_{\boldsymbol{x}} \mathcal{L}(\boldsymbol{f_\theta}(\boldsymbol{x}), y)||_2$. Similarly, Moosavi-Dezfooli et al. (2019) propose CURE: the regularization term $||\nabla_{\boldsymbol{x}} \mathcal{L}(\boldsymbol{f_\theta}(\boldsymbol{x}), y) - \nabla_{\boldsymbol{x}} \mathcal{L}(\boldsymbol{f_\theta}(\boldsymbol{x} + \boldsymbol{\delta}_{\text{FGSM}}), y)||_2$. Similarly to our analysis in Proposition 1, Moosavi-Dezfooli et al. (2019) relate their regularization term to a Finite Differences approximation of curvature. The regularization term in GradAlign is given by:

$$1 - \cos \left[ \nabla_{\boldsymbol{x}} \mathcal{L}(\boldsymbol{f_\theta}(\boldsymbol{x}), y), \; \nabla_{\boldsymbol{x}} \mathcal{L}(\boldsymbol{f_\theta}(\boldsymbol{x} + \boldsymbol{\eta}), y) \right], \quad \boldsymbol{\eta} \sim \text{Unif}[-\epsilon, \epsilon], \quad (5)$$

where $\cos[\boldsymbol{u}, \boldsymbol{v}] = \frac{\boldsymbol{u}^\top \boldsymbol{v}}{\sqrt{\boldsymbol{u}^\top \boldsymbol{u} \cdot \boldsymbol{v}^\top \boldsymbol{v}}}$ is the cosine similarity. Sriramanan et al. (2020) propose GAT: regularizing $||\boldsymbol{f_\theta}(\boldsymbol{x}) - \boldsymbol{f_\theta}(\boldsymbol{x} + \boldsymbol{\eta})||_2^2$ with $\boldsymbol{\eta} \sim \text{Bern}([-\epsilon, \epsilon]^d)$, where Bern is denoted as the Bernoulli distribution. Sriramanan et al. (2021) introduce NuAT: regularizing $||\boldsymbol{f_\theta}(\boldsymbol{x}) - \boldsymbol{f_\theta}(\boldsymbol{x} + \boldsymbol{\eta})||_*$, $\boldsymbol{\eta} \sim \text{Bern}([-\epsilon, \epsilon]^d)$. The GAT and NuAT regularization terms can be thought of enforcing the network to be locally constant, a specific case of local linearity. Qin et al. (2019) propose LLR: solving $\gamma(\epsilon, \boldsymbol{x}) = \max_{||\boldsymbol{\delta}||_\infty \leq \epsilon} |\mathcal{L}(\boldsymbol{f_\theta}(\boldsymbol{x} + \boldsymbol{\delta}), y) - \mathcal{L}(\boldsymbol{f_\theta}(\boldsymbol{x}), y) - \boldsymbol{\delta}^\top \nabla_{\boldsymbol{x}} \mathcal{L}(\boldsymbol{f_\theta}(\boldsymbol{x}), y)|$, obtaining $\boldsymbol{\delta}_{LLR}$ and regularize $\lambda \gamma(\epsilon, \boldsymbol{x}) + \mu \cdot \boldsymbol{\delta}_{LLR}^\top \nabla_{\boldsymbol{x}} \mathcal{L}(\boldsymbol{f_\theta}(\boldsymbol{x}), y)$. In Proposition 3, a similar result as in Proposition 1 is proven for LLR. We argue all of these methods either involve differentiating input gradients ($\nabla_{\boldsymbol{x}}(\cdot)$) with respect to the network parameters and therefore suffer from *Double Backpropagation* or donnot avoid CO. Our method, ELLE and ELLE-A, do not suffer from this computational inefficiency and can as well avoid CO, being therefore a more efficient alternative.

## 4 EXPERIMENTS

In this section, we conduct a thorough validation of the proposed method: ELLE(-A). We introduce the experimental setup in Sec. 4.1. Firstly, we demonstrate the effectiveness of $E_{\text{lin}}$ in detecting and controlling CO in Sec. 4.2. Next, we compare the performance of ELLE(-A) and related methods enforcing local linearity in Sec. 4.3. In Sec. 4.4 we compare against single-step methods in CIFAR10/100 and SVHN over multiple $\epsilon$ and training schedules. In Sec. 4.5 we analyze the performance of other single-step methods when adding our regularization term. Lastly, in Sec. 4.6 we analyze the performance of our method in the ImageNet dataset.

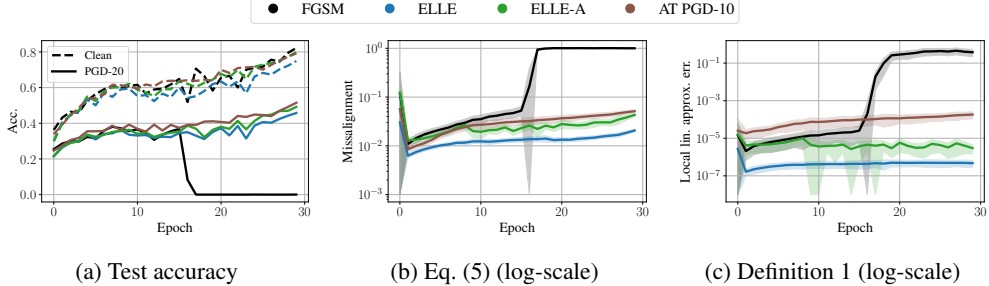

| (a) Test accuracy | (b) Eq. (5) (log-scale) | (c) Definition 1 (log-scale) |

Figure 2: Effectiveness of our local-linearity metric for detecting and controlling CO. We train with AT PGD-10, single step FGSM attacks and our method without (`ELLE`) and with (`ELLE-A`) adapting $\lambda$ at $\epsilon = 8/255$ in CIFAR10. We track: **(a)** the clean and PGD-20 test accuracies, **(b)** the GradAlign regularization term and **(c)** our regularization term. AT PGD-10 is able to produce locally linear models, see **(b), (c)**. Our regularization term accurately detects when CO appears and when regularized, is able to avoid CO. `ELLE-A` is able to attain a higher robustness than `ELLE`.

## 4.1 EXPERIMENTAL SETUP

We train the architectures PreActResNet18 (PRN), ResNet50 (He et al., 2016) and WideResNet-28-10 (WRN) (Zagoruyko and Komodakis, 2016) in CIFAR10/100 (Krizhevsky, 2009), SVHN (Netzer et al., 2011) and ImageNet (Deng et al., 2009). For CIFAR10 and SVHN, we train and test with $\epsilon$ values up to $26/255$ and $12/255$ respectively, as suggested by Yang et al. (2020) for the minimum Test-Train separation. For CIFAR100 and ImageNet we train up to $\epsilon = 26/255$ and $\epsilon = 8/255$ respectively. We use the SGD optimizer with momentum $0.9$ and weight decay $5 \cdot 10^{-4}$.

We are interested in which $\epsilon$ values and training schedules lead to CO. We coin the term *Delayed CO*.

**Observation 1** (Delayed Catastrophic Overfitting). *A training method suffers from Delayed CO if one of the following phenomenons is observed:*

- *(Perturbation-wise) With the same scheduler, CO is avoided for a perturbation size $\epsilon_1$, but not for a bigger one $\epsilon_2 > \epsilon_1$.*
- *(Duration-wise) With the same perturbation size $\epsilon$, CO is avoided for an scheduler of $L_1$ epochs, but not for an scheduler of $L_2 > L_1$ epochs.*

Our observation provides a name to the previously observed behavior in methods like FGSM AT, Free AT, Fast AT, GAT or NuAT (Andriushchenko and Flammarion, 2020; Li et al., 2020; Kim et al., 2021; de Jorge et al., 2022).To assess the appearance of *Delayed CO*, we distinguish between two main training schedules:

- *Short*: From Andriushchenko and Flammarion (2020), with 30 and 15 epochs for CIFAR10/100 and SVHN respectively, batch size of 128 and a cyclic learning rate schedule with a maximum learning rate of $0.2$.
- *Long*: From Rice et al. (2020), with 200 epochs, batch size of 128, a constant learning rate of $0.1$ for CIFAR10/100 and $0.01$ for SVHN, decayed by a factor of 10 at epochs 100 and 150.

For ImageNet, we use the 15-epoch schedule in Wong et al. (2020). The $\lambda$ selection for our method and its variants is done with a simple grid-search and its deferred to Appx. B. We evaluate the adversarial accuracy with PGD-20 and AutoAttack (AA) (Croce and Hein, 2020a) in the $\ell_\infty$ norm. To avoid unnecessary overhead, we remove the $1/255$ factor in $\epsilon$ in our tables and figures. In our tables, we highlight in red the experiments where CO appeared for at least one run and mark the best method and the runner-up in **bold** and underlined respectively. ImageNet experiments were conducted in a single machine with an NVIDIA A100 SXM4 80GB GPU. For the rest of experiments we used a single machine with an NVIDIA A100 SXM4 40 GB GPU.

## 4.2 LOCAL, LINEAR APPROXIMATION ERROR DETECTS AND CONTROLS CO

We firstly analyze the effectiveness of using Definition 1 as a regularization term for controlling CO. To do so, two questions should be answered: *Does the local linear approximation error suddenly*

*increase when CO appears? Does CO disappear when regularizing the local, linear approximation error?* In the following we answer both questions affirmatively.

We train a PRN on CIFAR10 with the *Short* schedule and FGSM attacks at $\epsilon = 8/255$, a setup where CO appears (Wong et al., 2020). We put AT PGD-10 results as a reference. Additionally, we train with our method, `ELLE`, and its adaptive variant, `ELLE-A` with $\lambda = 5,000$. We compare the value of Eqs. (2) and (5) and the clean and PGD-20 test accuracies per epoch. As expected and originally analyzed by Andriushchenko and Flammarion (2020), AT PGD-10 maintains a low local curvature of the loss during training.

**CO detection and avoidance with `ELLE`:** In Fig. 2, we observe that for FGSM training, our local, linear approximation error correlates with the gradient misalignment (Eq. (5)) spikes and the PGD-20 test accuracy drops. This verifies that the local, linear approximation error is a good CO detector. Additionally, when training with `ELLE` and `ELLE-A`, both the gradient misalignment and the local, linear approximation error remain close to zero, i.e., the network remains locally linear. Moreover, PGD-20 test accuracy follows a stable growth across epochs.

## 4.3 COMPARISON AGAINST METHODS ENFORCING LOCAL LINEARITY

In this section we compare `ELLE(-A)` against GradAlign, CURE and LLR. To avoid the expensive computation of $\gamma$ in LLR, we regularize $\left( \mathcal{L}(f_\theta(x + \delta), y) - \mathcal{L}(f_\theta(x), y) - \delta^\top \nabla_x \mathcal{L}(f_\theta(x), y) \right)^2$ and randomly sample $\delta \sim \text{Unif}\left( [-\epsilon, \epsilon]^d \right)$, a similar approach as in GradAlign and our method (Algorithm 1). We train PRN on CIFAR10 with the *Short* schedule at $\epsilon = 8/255$ and measure the clean accuracy, AA accuracy and total training time.

**Faster local linearity regularization:** In Fig. 1a we observe `ELLE(-A)` adds very little overhead to FGSM training, i.e., in total 5.06 min with FGSM v.s. 6.29 min with `ELLE(-A)` and is considerably faster than other methods enforcing local linearity such as GradAlign, with 15.89 min. In Appx. B.7 the runtime of the forward and backward passes is analyzed separately, showing the increased backward time in LLR, CURE and GradAlign due to *Double Backpropagation*. In Fig. 1b, we can firstly observe that our adaptive regularization variant (`ELLE-A`) greatly increases the performance when compared to `ELLE` at $\epsilon = 8/255$, e.g., $42.78\%$ v.s. $44.32\%$ AA accuracy. This performance boost can also be observed in Fig. 2. `ELLE` and `ELLE-A` match the performance of other methods at $\epsilon = 8/255$, e.g. $44.66\%$ and $44.32\%$ for GradAlign and `ELLE-A` respectively and for $\epsilon = 16/255$, e.g. $18.25\%$ and $18.28\%$ for CURE and `ELLE` respectively.

## 4.4 COMPARISON AGAINST SINGLE-STEP METHODS

Next, we analyze the performance of `ELLE` in comparison to state-of-the-art single-step AT algorithms. We train PRN networks with N-FGSM, GradAlign, SLAT (Park and Lee, 2021), Fast BAT, LLR, `ELLE-A` and `ELLE-A` jointly with the data augmentation scheme in N-FGSM, i.e., N-FGSM+`ELLE-A`. The expensive but reliable AT PGD-10 (Madry et al., 2018) is plotted as a reference in the short schedule. We use the *Short* schedule and the *Long* schedule. We use the recommended hyperparameters for each method. We average the performance over 3 random seeds.

***Short* schedule benchmark:** In Fig. 3 we can observe SLAT suffers from CO for $\epsilon \geq 4/255$ and $\epsilon \geq 10/255$ for SVHN and CIFAR10 respectively. GradAlign attains sub-optimal performance for $\epsilon > 16/255$ in CIFAR10 and behaves erratically in CIFAR100. Only the methods involving our regularization term are able to overcome CO for all $\epsilon$ and datasets. Note that N-FGSM suffers from CO for $\epsilon > 18/255$ in CIFAR100. Further, N-FGSM+`ELLE` consistently attains the best performance. It should be possible to attain similar performance to `ELLE(-A)` with GradAlign, but, we could not find a hyperparameter selection attaining matching `ELLE-A` for large $\epsilon$. This tuning difficulty was also reported by de Jorge et al. (2022). We report additional results showing the sensibility of GradAlign to hyperparameter changes in Appx. B.8. Overall, our regularization term contributes to closing the performance gap between single-step methods and the multi-step reference AT PGD-10.

***Long* schedule benchmark:** In Figs. 4a and 4b N-FGSM is affected from CO for large $\epsilon$, i.e., $\epsilon \geq 8/255$ and $\epsilon \geq 18/255$ for SVHN and CIFAR10 respectively. Fast-BAT is also affected by CO for all $\epsilon$ in CIFAR10 except $\epsilon \in \{1, 2, 10\}/255$. Contrarily, `ELLE` does not suffer from CO in any setup. Additionally, when adding our regularization term to N-FGSM (N-FGSM+`ELLE-A`), CO is

Table 1: **ImageNet results:** We report the PGD-50-10 and clean test accuracies. `ELLE-A` helps avoiding CO and when combined with N-FGSM provides the best performance at $\epsilon = 8/255$.

| $\epsilon$ | 2 | | 8 | |
|---|---|---|---|---|
| Method | PGD-50-10 | Clean | PGD-50-10 | Clean |
| FGSM | $42.61 \pm (0.13)$ | $63.66 \pm (0.07)$ | $5.23 \pm (4.15)$ | $48.85 \pm (4.70)$ |
| N-FGSM | $\mathbf{44.54} \pm (0.30)$ | $61.11 \pm (0.17)$ | $\underline{13.32} \pm (1.21)$ | $43.29 \pm (1.41)$ |
| `ELLE-A` | $41.49 \pm (0.39)$ | $62.05 \pm (0.50)$ | $12.40 \pm (1.78)$ | $42.40 \pm (6.36)$ |
| N-FGSM+`ELLE-A` | $\underline{42.20} \pm (0.52)$ | $58.95 \pm (0.43)$ | $\mathbf{14.58} \pm (1.10)$ | $38.04 \pm (3.31)$ |

avoided for all $\epsilon$ in both SVHN and CIFAR10. We identify `ELLE-A` attains a lower final PGD-20 accuracy than `ELLE` due to robust overfitting. To complete the analysis, we train a WRN architecture in CIFAR10 with $\epsilon \in \{8/255, 16/255, 26/255\}$, observing that in the *Long* schedule, N-FGSM suffered from CO for $\epsilon = 26/255$ while `ELLE` remained resistant, see Fig. 4c.

## 4.5 REGULARIZING OTHER SINGLE-STEP METHODS

In this section we study the performance of N-FGSM and GAT (Sriramanan et al., 2020) when combined with `ELLE-A`. We train PRN and WRN architectures in the CIRFAR10 dataset at $\epsilon \in \{16, 26\}/255$ and report the clean and AA accuracies. In Fig. 5 we observe the performance of GAT and N-FGSM is matched or increased in every setup, specially in the cases where CO appeared. Noticeably, the performance increased from $20.54\%$ to $21.28\%$ with WRN and N-FGSM at $\epsilon = 16/255$ and from $10.96\%$ to $12.03\%$ with PRN and N-FGSM at $\epsilon = 26/255$. Interestingly, there was one case where simply changing the architecture from PRN to WRN made CO appear, i.e., N-FGSM at $\epsilon = 26/255$. This suggests architecture might play a role in the appearance of CO. Nevertheless, when adding `ELLE-A`, CO was avoided in every setup. This favours the addition of `ELLE-A` regularization when the appearance of CO is unsure or performance in large $\epsilon$ is crucial.

## 4.6 IMAGENET RESULTS

We train ResNet50 with the training setup of Wong et al. (2020) with the only difference that as de Jorge et al. (2022), we consider a maximum image resolution of $288 \times 288$. We compare the PGD-50-10 adversarial accuracy when training with FGSM, N-FGSM and `ELLE-A`.

In Table 1 we observe that FGSM suffers from CO for $\epsilon = 8/255$, while `ELLE-A` does not suffer from CO for any $\epsilon$. Moreover, our method improves the performance at $\epsilon = 8/255$ when in combination with N-FGSM, i.e., from $13.32\%$ to $14.58\%$ PGD-50-10 accuracy.

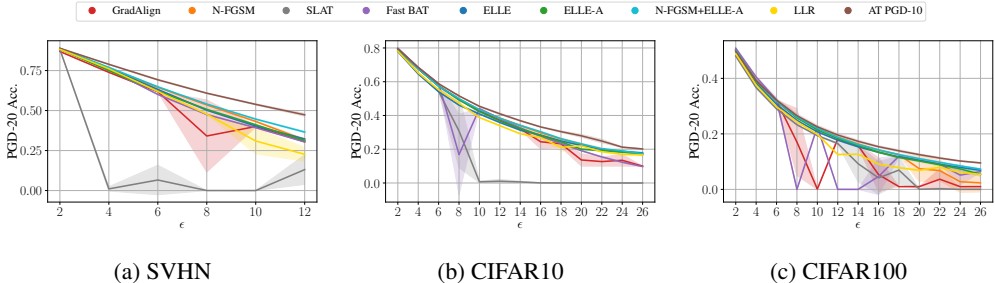

(a) SVHN  (b) CIFAR10  (c) CIFAR100

Figure 3: **Catastrophic Overfitting in the *Short* schedule:** Comparison of our method against single-step methods and AT PGD-10 on **(a)** SVHN, **(b)** CIFAR10 and **(c)** CIFAR100. `ELLE-A` and N-FGSM+`ELLE-A` are the only single-step methods avoiding CO while attaining high performance for all $\epsilon$ and datasets.

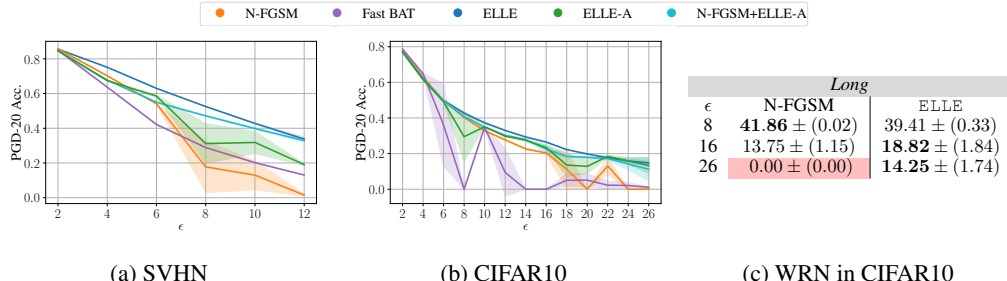

Figure 4: **Catastrophic Overfitting in the *Long* schedule:** We report the PGD-20 adversarial accuracy for the PRN architecture in (a) SVHN and (b) CIFAR10. In **(c)** we report the PGD-20 adversarial accuracy for the WRN architecture trained in CIFAR10 with the *Long* schedule. N-FGSM suffers from CO in both SVHN and CIFAR10 datasets for $\epsilon > 6/255$ and $\epsilon > 16/255$ respectively. `ELLE` remains resistant to CO in all setups.

## 5 CONCLUSION

We propose a cost-effective regularization term that can prevent catastrophic overfitting (CO) in single-step adversarial training (AT) by enforcing local linearity. We establish a relationship between our regularization term and second-order directional derivatives, demonstrating that the proposed regularization implicitly smooths the loss landscape. Our regularization term can detect CO and is more efficient in avoiding CO than GradAlign and other methods enforcing local linearity. We conduct a thorough evaluation of our method with large adversarial perturbations, long training schedules and in combination with other single-step methods. Our regularization term is a simple plug-in useful when the appearance of CO is unsure and to improve the performance in large $\epsilon$.

**Limitations:** The main limitation of our method is the need to run a forward pass on three extra points per training sample, resulting in additional memory usage. Previous methods enforcing local linearity only need one extra point per training sample but incurr in a $\times 3$ runtime (Simon-Gabriel et al., 2019; Ross and Doshi-Velez, 2018; Moosavi-Dezfooli et al., 2019; Qin et al., 2019). Our method is considerably faster overall as we observe in Fig. 1. We leave the search of more memory efficient methods for enforcing local linearity as future work.

**Future directions:** Our method is able to overcome CO and greatly improve the performance in large $\epsilon$ scenarios, see Fig. 3. Large $\epsilon$ values have been sparsely utilized before and generalization is attained (Andriushchenko and Flammarion, 2020; de Jorge et al., 2022). However, whether a classifier should be robust for a certain image $\boldsymbol{x}$ and a (large) perturbation budget $\epsilon$ is unknown and requires human intervention. Addepalli et al. (2022) find attacking images with low contrast with large $\epsilon$ can lead in a flip in the oracle's prediction. Yang et al. (2020) provide upper bounds up to which $100\%$ adversarial accuracy is theoretically achievable, nevertheless, these bounds might not be realistic for a human oracle. In Appx. B.1 we display large $\epsilon$ attacks and find no perturbation clearly changes the human oracle prediction. We believe though that studying the $\epsilon$ upper bounds is needed for advancing the CO and AT fields.

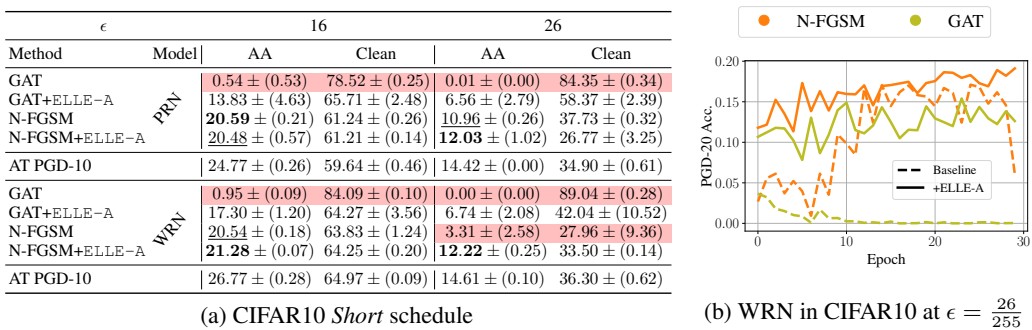

Figure 5: **Combining N-FGSM and GAT with** `ELLE-A`**: (a)** AutoAttack (AA) and Clean accuracy for PRN and WRN trained with $\epsilon \in \{16, 26\}/255$. **(b)** Evolution of PGD-20 test accuracy during training of WRN. `ELLE-A` helps GAT and N-FGSM overcome CO and improve their performance.

## ACKNOWLEDGEMENTS

Authors acknowledge the constructive feedback of reviewers and the work of ICLR'24 program and area chairs. We thank Zulip[1] for their project organization tool. ARO - Research was sponsored by the Army Research Office and was accomplished under Grant Number W911NF-24-1-0048. Hasler AI - This work was supported by Hasler Foundation Program: Hasler Responsible AI (project number 21043) SNF project – Deep Optimisation - This work was supported by the Swiss National Science Foundation (SNSF) under grant number 200021_205011. Fanghui Liu is supported by the Alan Turing Institute under the UK-Italy Trustworthy AI Visiting Researcher Programme. Pablo M. Olmos acknowledges the support by the Spanish government MCIN/AEI/10.13039/501100011033/ FEDER, UE, under grant PID2021-123182OB-I00, and by Comunidad de Madrid under grants IND2022/TIC-23550 and ELLIS Unit Madrid.

## REPRODUCIBILITY STATEMENT

In the course of this project, we strictly employ publicly available benchmarks, ensuring the reproducibility of our experiments. We report the pseudo-code of our implementation in Algorithm 1. Furthermore, we furnish extensive details regarding the hyperparameters utilized in our investigation and endeavor to provide comprehensive insights into all the techniques employed. Our implementation is publicly available in `https://github.com/LIONS-EPFL/ELLE`.

## ETHICS STATEMENT

Authors acknowledge that they have read and adhere to the code of ethics.

## BROADER IMPACT

The robustness of neural networks to (worst-case) attacks is of crucial interest to the community. Given that neural networks are increasingly deployed in real-world applications, avoiding attacks is important. At the same time, obtaining robust classifiers at this point is very costly, since cheap alternatives suffer from catastrophic overfitting (CO). Therefore, we do believe that `ELLE` can have a positive impact in machine learning. Concretely, the improvement we show through avoiding CO contributes to the development of more reliable and trustworthy Machine Learning systems. Additionally, the efficiency of our method reduces the computational requirements for training robust models, making them more accessible to a wider range of researchers and practitioners. We hope this can help democratize the entrance barrier to the area and foster further advances in AT. However, we encourage the community to analyze further the potential negative impacts from such methods on robustness.

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

CONTENTS OF THE APPENDIX

In Appx. A we briefly introduce the objectives of robust training and relevant works in the area. We introduce additional experiments in Appx. B. Finally, in Appx. C we include our proofs and additional theoretical connections.

## A    ROBUST TRAINING

In standard training, we minimize the Cross-Entropy loss $\mathcal{L}$ evaluated at our training samples, i.e., solving $\min_{\boldsymbol{\theta}} \mathbb{E}_{(\boldsymbol{x},y)\sim\mathcal{D}} [\mathcal{L}(\boldsymbol{f_\theta}(\boldsymbol{x}), y)]$, where $\boldsymbol{\theta}$ are the parameters of our classifier $f$ and $\mathcal{D}$ is the distribution of our training data. In contrast, in AT (Madry et al., 2018), we minimize the worst case of the loss under bounded adversarial perturbations:

$$\min_{\boldsymbol{\theta}} \mathbb{E}_{(\boldsymbol{x},y)\sim\mathcal{D}} \left[ \max_{||\boldsymbol{\delta}||_p \leq \epsilon} \mathcal{L}(\boldsymbol{f_\theta}(\boldsymbol{x} + \boldsymbol{\delta}), y) \right] , \qquad \text{(AT)}$$

where $1 \leq p \leq \infty$. AT has notably withstood the test of time, proving to be resistant to powerful adversarial attacks (Croce and Hein, 2020b). Since its appearance, several other robust training methods have been proposed such as TRADES (Zhang et al., 2019) or DyART (Xu et al., 2023). Additionally, several concerns regarding AT (and other robust training methods) have been pointed out, such as its excessive increase in the separation margin (Rade and Moosavi-Dezfooli, 2022) or its computational inefficiency (Shafahi et al., 2019).

## B    ADDITIONAL EXPERIMENTAL VALIDATION

We visualize attacks towards our networks in Appx. B.1. The hyperparameter selection for our methods for all experiments in the main paper is included in Appx. B.2. We study possible variations of our methods in Appx. B.3. We include variations of the *Long* scheduler in Appendices B.5 and B.6. Models from Fig. 3 are evaluated with AutoAttack in Appx. B.9. Various ablation studies are available in Appendices B.8, B.10 and B.11.

### B.1    VISUALIZATION OF LARGE-$\epsilon$ ATTACKS

Previous works sparsely consider training with large $\epsilon$ for CIFAR10 but donnot visualize such attacks (Andriushchenko and Flammarion, 2020; de Jorge et al., 2022). Addepalli et al. (2022) analyze adversarial attacks and defences under large $\epsilon$ and argue at $\epsilon \geq 24/255$ it is possible to swith the oracle prediction for low contrast images. For ImageNet, Qin et al. (2019) consider $\epsilon = 16/255$ and visualize a single image arguing it is difficult to recognize the object after the attack. We visualize PGD-50 and FGSM attacks in 9 randomly sampled images from the test sets of CIFAR10/100 and ImageNet at $\epsilon = 26/255$, $\epsilon = 26/255$ and $\epsilon = 16/255$ respectively.

In Fig. 6 we can observe the images after the adversarial attacks present visible perturbations, but donnot clearly change the oracle prediction. As argued by Addepalli et al. (2022) and in Sec. 5, the ability of the attacker to flip the oracle prediction might vary from image to image and needs to be further studied in the future. Nevertheless, we have strong reasons to believe a high robustness can still be attained for large $\epsilon$, specially for the higher resolution ImageNet.

### B.2    $\lambda$ SELECTION

In this section, we study how to select the appropriate regularization parameter for `ELLE(-A)` under different schedulers and datasets. We evaluate the PGD-20 adversarial accuracy in a 1024-image validation sample extracted from the training set of each dataset.

`ELLE`: We test the value for the regularization term in our vanilla method (`ELLE`) in CIFAR10 and SVHN. We train PRN in the *Short* schedule with a wide range of $\lambda$ values for all $\epsilon$ values, i.e.:

$$\lambda_{\text{SVHN}} \in \{2 \cdot 10^2, 2 \cdot 10^3, 10^4, 2 \cdot 10^4, 2 \cdot 10^5\}$$
$$\lambda_{\text{CIFAR10}} \in \{10^2, 10^3, 5 \cdot 10^3, 10^4, 10^5\}$$

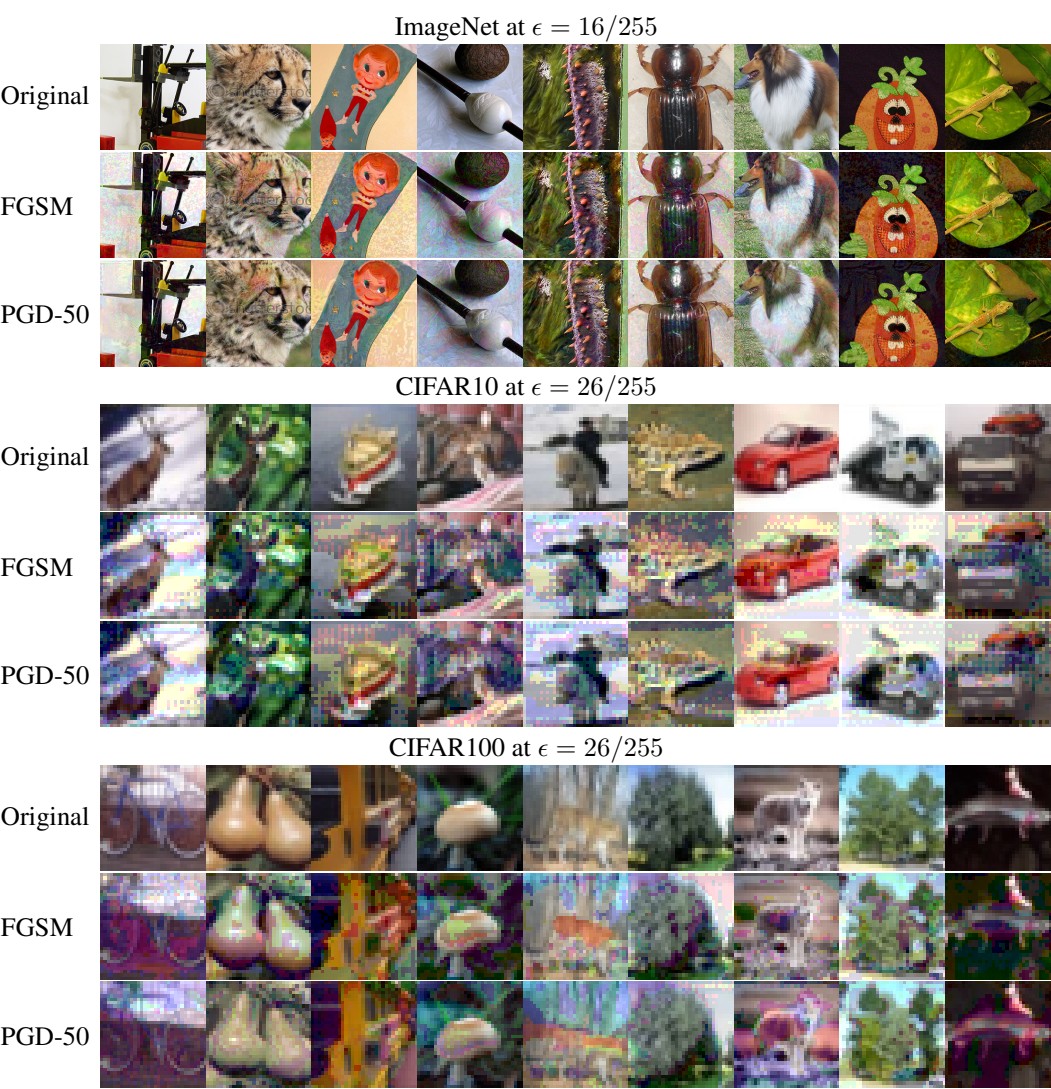

Figure 6: Adversarial perturbations for CIFAR10/100 at $\epsilon = 26/255$ and ImageNet at $\epsilon = 16/255$ with FGSM and PGD-50 attacks. Original image displayed as a reference. Perturbations are visible but donnot clearly affect the human prediction, specially for high resolution images in ImageNet.

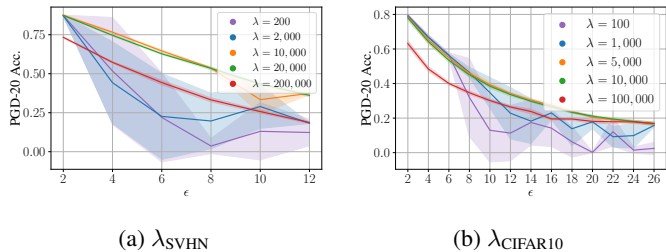

(a) $\lambda_{\text{SVHN}}$        (b) $\lambda_{\text{CIFAR10}}$

Figure 7: Grid search for $\lambda$ in SVHN and CIFAR10 datasets with `ELLE`. `ELLE` suffers from CO for $\lambda_{\text{SVHN}} \in \{200;\ 2,000;\ 10,000\}$ and $\lambda_{\text{CIFAR10}} \in \{100;\ 1,000\}$.

Note that we choose values of $\lambda_{\text{SVHN}}$ double of those of $\lambda_{\text{CIFAR10}}$, we follow the results of Andriushchenko and Flammarion (2020), in which this proportionality is observed. In Figs. 7a and 7b, we find that for smaller $\lambda$ values, CO still appears, e.g., $\lambda_{\text{SVHN}} \in \{200;\ 2,000;\ 10,000\}$ and $\lambda$CIFAR10 $\in \{100;\ 1,000\}$. When $\lambda$ is sufficiently large, CO is avoided but there is a performance degradation, this is clearly observed for $\lambda_{\text{SVHN}} = 200,000$ and $\lambda_{\text{CIFAR10}} = 100,000$, where CO is not observed but PGD-20 accuracy is decreased for smaller $\epsilon$. To avoid an expensive grid search, we simply use $\lambda_{\text{CIFAR100}} = 5,000$ since it provides a good performance for all values of $\epsilon$ in CIFAR10. In the *Long* schedule, to make sure CO is avoided and avoid an expensive grid search, we reutilize the largest values with a good performance from the *Short* schedule, i.e., $\lambda_{\text{SVHN}} = 20,000$ and $\lambda_{\text{CIFAR10}} = 10,000$.

`ELLE-A`: In the case of adaptive $\lambda$, we tune the value of $\lambda_{\max}$ in Algorithm 1. We take advantage of the results from Fig. 7 and interpolate between $\lambda$ parameters close to the best performing ones in `ELLE`. In particular we test $\lambda_{\text{SVHN}} \in \{5,000;\ 10,000;\ 15,000;\ 20,000\}$ and $\lambda_{\text{CIFAR10}} \in \{1,000;\ 2,000;\ 3,000;\ 4,000;\ 5,000\}$. We observe that for $\lambda_{\text{SVHN}} > 5,000$, all methods behave similarly except for $\epsilon \geq 10/255$, where larger $\lambda_{\text{SVHN}}$ improved the performance. Similarly, for $\lambda_{\text{CIFAR10}} \geq 1,000$, the performance is almost the same for all $\epsilon$, therefore, we take the $\lambda_{\text{CIFAR10}}$ with the lowest average standard deviation across $\epsilon$, i.e., $\lambda_{\text{CIFAR10}} = 4,000$ with an average standard deviation of 0.008. Similarly as for `ELLE`, for CIFAR100 we simply use $\lambda_{\text{CIFAR100}} = \lambda_{\text{CIFAR10}} = 4,000$ and for the *Long* schedule, we use $\lambda_{\text{SVHN}} = 20,000$ and $\lambda_{\text{CIFAR10}} = 10,000$. For ImageNet we choose $\lambda = 10,000$.

**Combination with other methods:** Based on previous results for `ELLE` and `ELLE-A`, for the results in Sec. 4.5 we use $\lambda = 4,000$ for N-FGSM in both PRN and WRN. For GAT, a larger regularization was required, we used $\lambda = 10,000$ and $\lambda = 50,000$ for PRN-18 and WRN respectively. For N-FGSM+`ELLE-A` in the SVHN dataset we use $\lambda = 5,000$ and for CIFAR100, $\lambda = 4,000$. In the case of N-FGSM+`ELLE-A` on ImageNet, we choose $\lambda = 20,000$ for $\epsilon \in \{2,8\}/255$ and $\lambda = 100,000$ for $\epsilon = 16/255$. A higher performance could be attained by further fine tuning the regularization parameter.

**General guideline:** Based on our experiments, we observe that values of $\lambda$ in the range $[4,000;\ 20,000]$ tend to avoid CO and provide the best performance for all datasets.

## B.3   ABLATION STUDIES

We test the effectiveness of using other local linearity metrics:

- **Two triplets:** `ELLE(-A)` estimates the expectation in Definition 1 with a single $(\boldsymbol{x}_a, \boldsymbol{x}_b, \alpha)$ triplet sample, we test how approximating the expectation with two samples affects the metric.

- $\alpha = 0.5$**:** In Definition 1, $\alpha$ is sampled uniformly from the $[0, 1]$ interval. We analyze the effect of fixing $\alpha = 0.5$.

- $\alpha_{\textbf{max}}$**:** We test the effect of choosing
$$\alpha_{\max} = \arg\max\nolimits_{\alpha \in [0,1]} \left| \mathcal{L}(\boldsymbol{f}_{\boldsymbol{\theta}}(\boldsymbol{x}_c), y^i) - (1 - \alpha) \cdot \mathcal{L}(\boldsymbol{f}_{\boldsymbol{\theta}}(\boldsymbol{x}_a), y^i) - \alpha \cdot \mathcal{L}(\boldsymbol{f}_{\boldsymbol{\theta}}(\boldsymbol{x}_b), y^i) \right|^2,$$

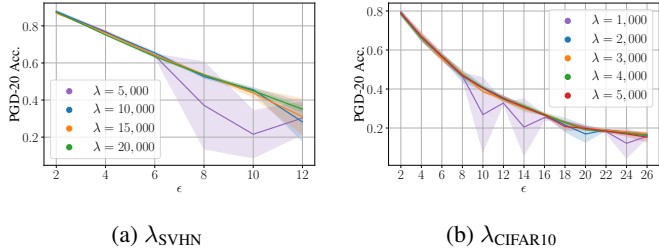

(a) $\lambda_{\text{SVHN}}$             (b) $\lambda_{\text{CIFAR10}}$

Figure 8: Grid search for $\lambda$ in SVHN and CIFAR10 datasets with ELLE-A. ELLE-A suffers from CO for $\lambda_{\text{SVHN}} = 5,000$ and $\lambda_{\text{CIFAR10}} = 1,000$. We select $\lambda_{\text{SVHN}} = 20,000$ and $\lambda_{\text{CIFAR10}} = 4,000$ for the rest of our experiments.

i.e., the value that maximizes the local, linear approximation error for a given $(\boldsymbol{x}_a, \boldsymbol{x}_b)$-sample. We obtain $\alpha_{\text{max}}$ with a PGD-10 procedure with a step-size of 0.1.

We analyze the value of these metrics when training with FGSM in CIFAR10 with $\epsilon = 8/255$. This way we can observe behavior of the metrics both before and after CO.

**Local linearity metrics:** In Fig. 18a we can observe all the proposed metrics behave similarly before and after CO appears, only differing by a constant magnitude across all epochs. This immediately discards the use of the two memory expensive triplets and the expensive to compute $\alpha_{\text{max}}$. We argue using $\alpha = 0.5$ is a feasible approach and similar results would be obtained by appropriately changing the regularization parameter $\lambda$.

### B.4 ADDITIONAL ELLE-A RESULTS

In this section, we analyze in depth the mechanics of ELLE-A in terms of the $\lambda$ evolution along time, and the influence of its hyperparameter $\lambda_{\text{max}}$.

**Evolution of $\lambda$:** In Algorithm 1, we increase $\lambda$ to $\lambda_{\text{max}}$ when $E_{\text{lin}} > \text{mean}(\text{err\_list}) + 2 \cdot \text{std}(\text{err\_list})$. We track the value of $E_{\text{lin}}$, $C_{\text{lin}} = \text{mean}(\text{err\_list}) + 2 \cdot \text{std}(\text{err\_list})$ and $\lambda$ accross training steps for WRN trained on CIFAR10 at $\epsilon = \in \{8, 16, 26\}/255$ with the *Short* schedule and with ELLE-A and N-FGSM+ELLE-A. Also,

In Fig. 9 we observe $\lambda$ is increased more frequently as $\epsilon$ increases. Additionally, when using N-FGSM data augmentation, this frequency is reduced. This suggests our adaptive $\lambda$ scheme correctly captures when local linearity is suddenly increasing and accordingly corrects it.

### B.5 ADDITIONAL LONG SCHEDULE RESULTS

**Additional scheduler details:** In addition to the schedulers used in Sec. 4, we use the *Long-cos* scheduler of Xu et al. (2023), which is also 200 epochs long, but has a different scheduling and a batch size of 256. We include a visual comparison of *Short*, *Long* and *Long-cos* schedules in Fig. 10. Since the *Long-cos* is used in state-of-the-art AT methods such as DyArt, we are interested in analyzing the appearance of CO in this scenario.

**CO in the *Long* schedule:** In Fig. 11, we can observe the training curves for GradAlign, N-FGSM, ELLE, ELLE-A and N-FGSM+ELLE-A when trained with the *Long* schedule with $\epsilon = 18/255$. The evolution of the local linearity (Fig. 11b) and the PGD-20 test accuracy (Fig. 11a) denote that N-FGSM clearly suffers from CO in this setup, while GradAlign and ELLE(-A) do not. Similarly as in Fig. 2 for FGSM training, the network becomes highly non-linear and PGD-20 test accuracy drops to zero for N-FGSM from the 29th epoch onwards. When combining N-FGSM with our regularization term (N-FGSM+ELLE-A), CO is avoided.

**CO in the *Long-cos* schedule:** We train PRN with GradAlign, N-FGSM and ELLE on CIFAR10 with the *Long-cos* schedule and report the average PGD-20 accuracy over three runs. In Fig. 13, we observe N-FGSM suffers from Robust Overfitting for $\epsilon \in [8/255, 16/255]$, which is consistent with the observations of de Jorge et al. (2022) for long schedules. For $\epsilon > 16/255$, N-FGSM exhibits

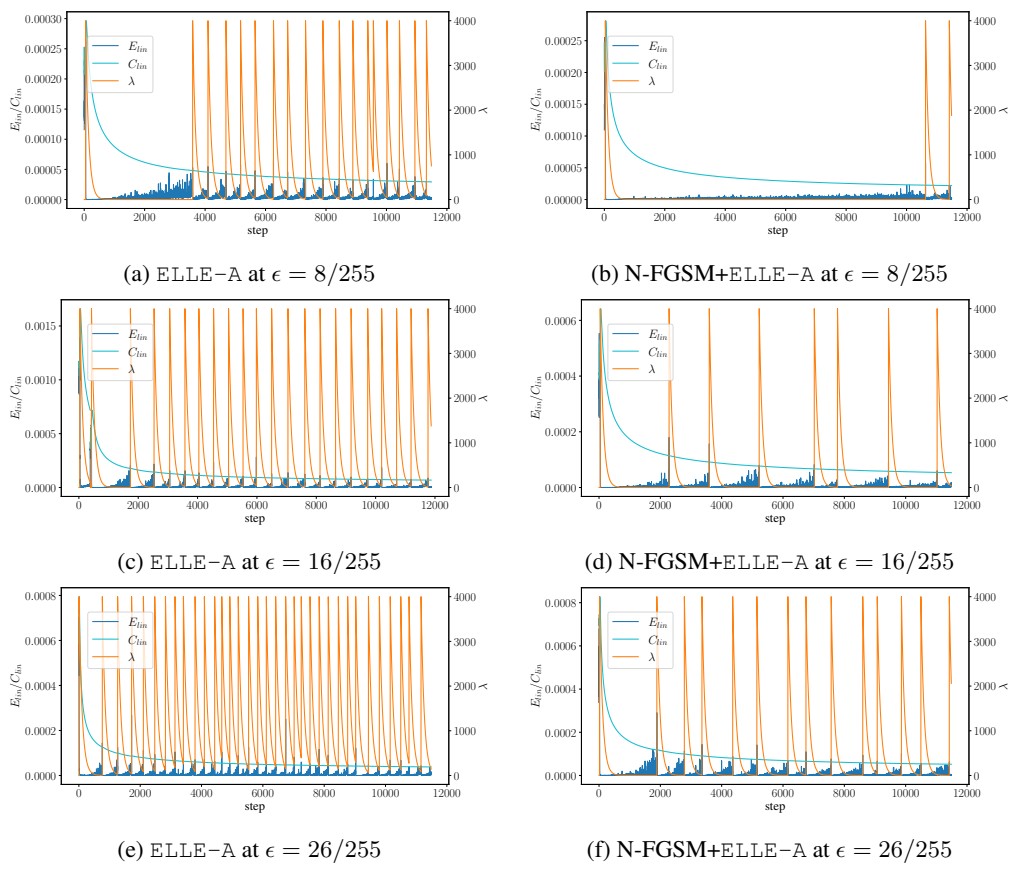

Figure 9: Evolution of `ELLE-A` and N-FGSM+`ELLE-A` during training for $\epsilon \in \{8, 16, 26\}/255$. The condition to increase $\lambda$ to $\lambda_{\max}$, i.e., $E_{\text{lin}} > C_{\text{lin}}$, is met more frequently as $\epsilon$ increases. Introducing N-FGSM data augmentation reduces the amount of times the condition is met, suggesting that data augmentation helps enforcing local linearity but it is not enough.

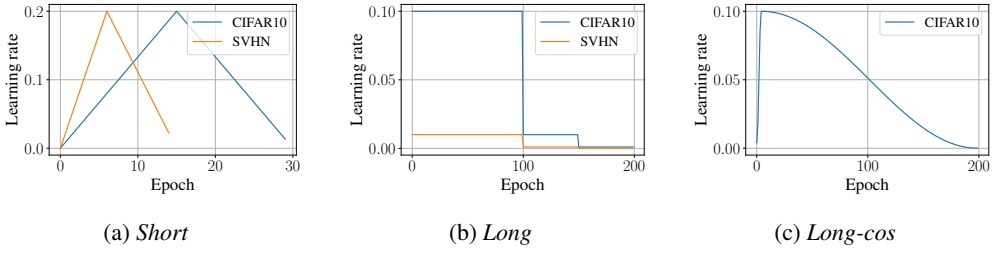

Figure 10: **Scheduler comparison:** Learning rate per epoch for the three schedules considered in this work. Note that in **(a)** the *Short* schedule for the SVHN dataset is only 15 epochs long.

CO, leading to low mean and high variance PGD-20 accuracy. In contrast, `ELLE` consistently avoids catastrophic overfitting for all $\epsilon$ values. Our analysis of GradAlign reveals that this method shows an erratic behavior, with catastrophic overfitting observed for some $\epsilon$ values, such as $\epsilon = 8/255$, $\epsilon = 16/255$, and $\epsilon = 18/255$. Additionally, GradAlign converges to a random classifier for $\epsilon = 24/255$ and $\epsilon = 26/255$. Our findings provide valuable insights into the behavior of these adversarial attack methods and highlight the importance of carefully analyzing the sensitivity to $\epsilon$ values. Additionally, in Fig. 12, we observe the evolution of the test accuracies and local linearity during training for $\epsilon = 18/255$. Similarly to Fig. 11, GradAlign and `ELLE` remain resistant to CO and are able to control local linearity. On the contrary, N-FGSM suffers from CO from the 150[th] epoch, where the network becomes highly non-linear and PGD-20 accuracy drops to 0.

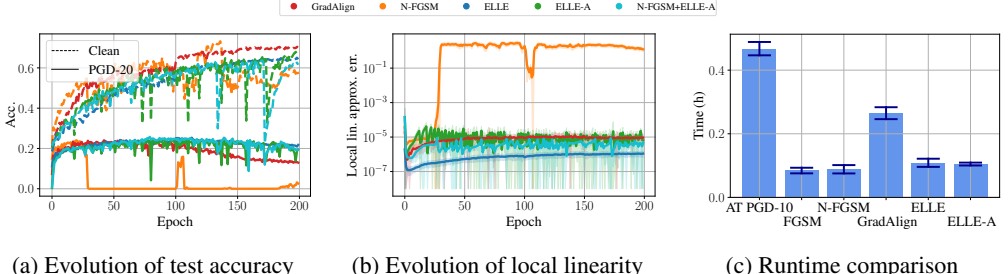

(a) Evolution of test accuracy        (b) Evolution of local linearity        (c) Runtime comparison

Figure 11: **Delayed CO (a)-(b):** CO in *Long* training schedule, CIFAR10 and $\epsilon = 18/255$. We track: **(a)** the clean and PGD-20 test accuracies and **(b)** our regularization term. GradAlign and ELLE, which explicitly enforce local linearity, do not suffer from CO. N-FGSM presents CO at the $30^{\text{th}}$ epoch, where the local linear approximation error spikes and the PGD-20 test accuracy drops to 0. **Efficient local linearity regularization (c):** Average runtime per training step for various methods. ELLE is capable of enforcing local linearity while adding little overhead to standard FGSM training. On the contrary, GradAlign has a heavy overhead due to *double backpropagation*.

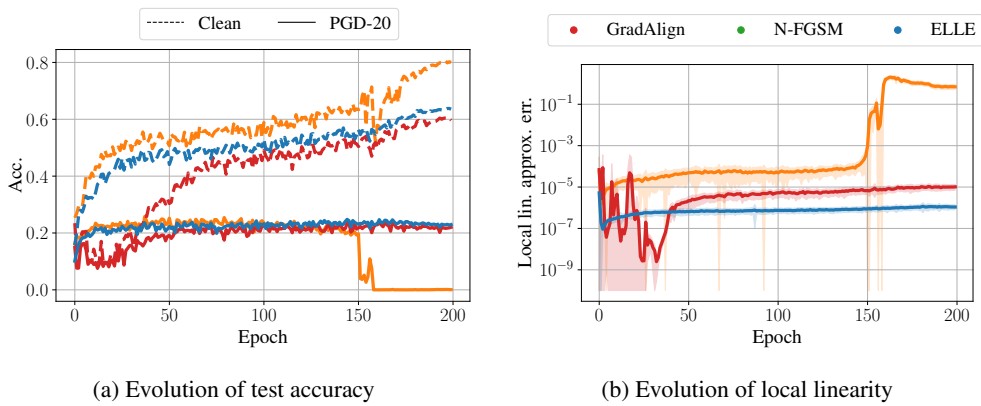

(a) Evolution of test accuracy                     (b) Evolution of local linearity

Figure 12: **Delayed CO (a)-(b):** CO in *Long-cos* training schedule and $\epsilon = 18/255$. We track **(a)** the clean and PGD-20 test accuracies (a) GradAlign (Andriushchenko and Flammarion, 2020) and ELLE, which explicitly enforce local linearity, do not suffer from CO. N-FGSM (de Jorge et al., 2022) presents CO at the $150^{\text{th}}$ epoch, where the local linear approximation error spikes and the PGD-20 test accuracy drops to 0.

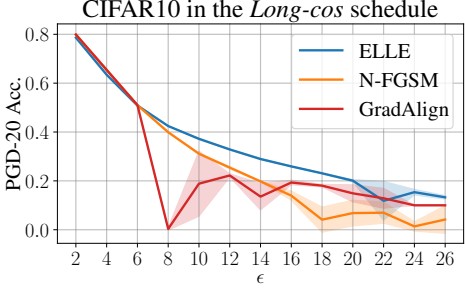

Figure 13: **Delayed CO in the *Long-cos* scheduler:** We report the PGD-20 accuracy for ELLE, N-FGSM and GradAlign for $\epsilon \in \{2/255, \cdots 26/255\}$. GradAlign does not avoid CO with the default regularization parameters with the *Long-cos* scheduler. N-FGSM suffers from CO for $\epsilon > 16/255$. ELLE remains resistant to CO.

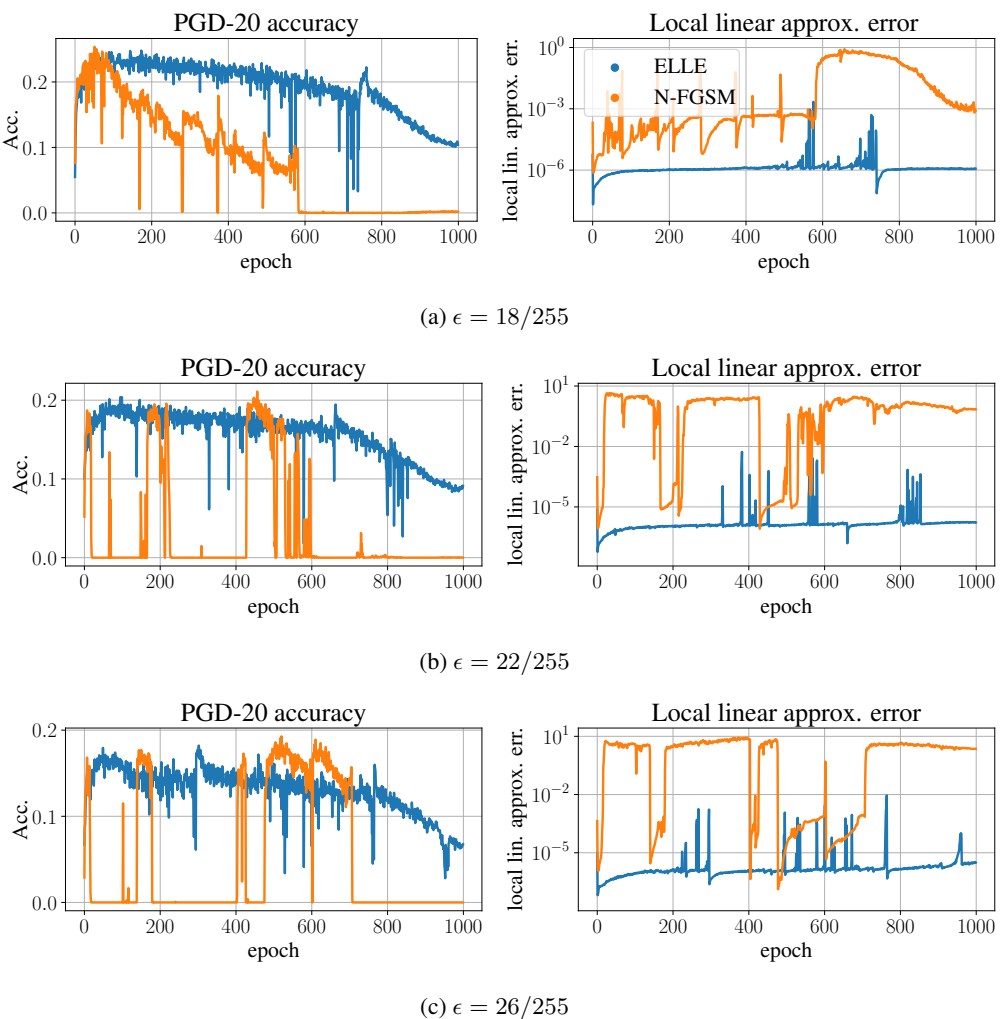

(a) $\epsilon = 18/255$

(b) $\epsilon = 22/255$

(c) $\epsilon = 26/255$

Figure 14: **1000-epoch long schedule:** CIFAR10 PGD-20 test accuracy and local linear approximation error evolution along 1000 epochs with the *Long-cos* schedule for `ELLE` and N-FGSM for $\epsilon \in \{18/255, 22/255, 26/255\}$ (**(a)**, **(b)**, and **(c)** respectively). N-FGSM presents CO in all scenarios, while `ELLE` only presents robust overfitting.

## B.6  1000-EPOCH LONG SCHEDULES

To demonstrate our method is capable of avoiding CO for even longer schedules, we test the performance of `ELLE` and N-FGSM for $\epsilon \in \{18/255, 22/255, 26/255\}$ when training the PreActResNet architecture with the cosine schedule, a maximum learning rate of $0.02$ and $1000$ epochs in CIFAR10.

In Fig. 14, we can observe that N-FGSM presents a chaotic behavior, where PGD-20 test accuracy is suddenly lost and gained within a few epochs. On the contrary, `ELLE` follows an homogeneous trend with no CO.

## B.7  RUNTIME COMPARISON

In order to better understand the computational advantages of using `ELLE`, we compare the average runtime per training step of AT PGD-10, FGSM, N-FGSM, LLR, CURE, GradAlign and `ELLE(-A)`. We report the runtime of the forward pass, of the backward pass and the total runtime per step. The PRN and WRN architectures are trained for $5$ epochs with a batch size of $128$ for this experiment.

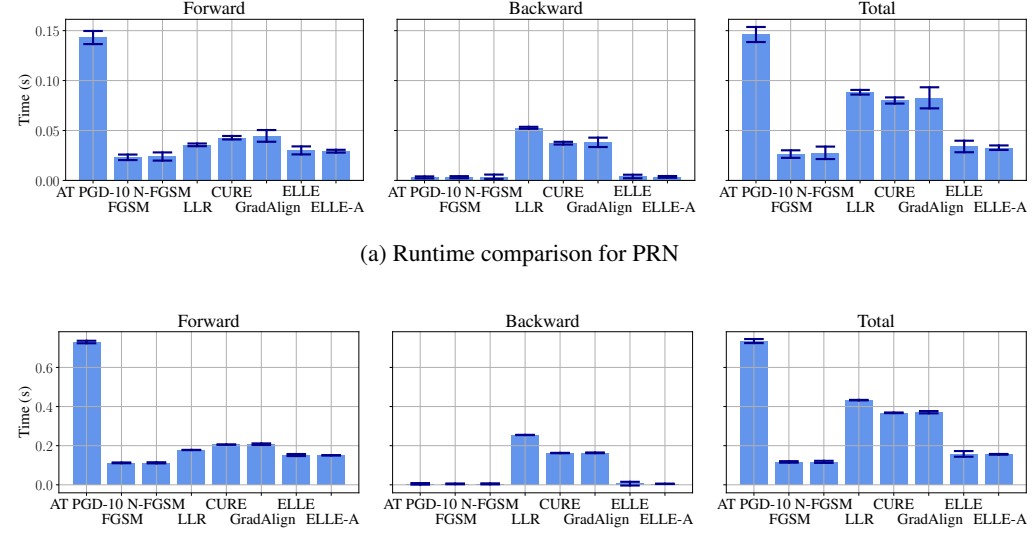

(a) Runtime comparison for PRN

(b) Runtime comparison for WRN

Figure 15: Forward, backward and total runtime in average per training step for the PRN **(a)** and WRN **(b)** architectures. FGSM, N-FGSM and ELLE(-A) attain similar total runtimes in both scenarios. LLR, CURE and GradAlign have a considerably larger backward runtime than the rest. This is due to double backpropagation. ELLE(-A) avoids double backpropagation and obtains similar backward runtimes to FGSM.

Two main insights can be drawn from Fig. 15: i) ELLE adds very little overhead to FGSM training and ii) LLR, CURE and GradAlign add an expensive overhead in the backward pass due to *Double Backpropagation*. ELLE, while based on enforcing local linearity as LLR, CURE and GradAlign, is considerably more efficient.

### B.8 GRADALIGN ABLATION

In this section we perform additional studies in the effect of the $\lambda$ regularization parameter in GradAlign. We train PRN architectures with the *Short* schedule in CIFAR10 with $\epsilon = 26/255$ and $\lambda_{\text{GradAlign}} \in \{0.25, 0.5, 1, 2, 4, 8, 16\}$ with three different random seeds. We report the performance of ELLE as a baseline.

**GradAlign is highly sensible to $\lambda_{\text{GradAlign}}$:** In Fig. 16a we find only for $\lambda_{\text{GradAlign}} = 1$, CO is avoided and the network does not converge to a constant classifier. Additionally, a suboptimal performance is obtained in comparison to ELLE, i.e. $15.90\%$ v.s. $17.80\%$. This shows the difficulty of choosing $\lambda_{\text{GradAlign}}$.

**GradAlign regularization term is unstable:** In Figs. 16b to 16d we report the PGD-20 test accuracy, the local, linear approximation error and the gradient missalignment (Eq. (5)) respectively for $\lambda_{\text{GradAlign}} = 8$. Since $\lambda_{\text{GradAlign}}$ is so large, we have that the network converges to a constant classifier with $10\%$ accuracy. Nevertheless, we find that the regularization term in GradAlign is inconsistent with this fact. As the model is constant, it is also linear and we should have that the gradient missalignment is close to $0$. Nevertheless this metric converges to $1$ during training. Alternatively, the local, linear approximation error is consistent with the convergence to a constant classifier and goes to $0$.

In conclusion, together with the findings in Appx. B.7 and Sec. 4.2. We find that ELLE is a more consistent, efficient and easier to tune method than GradAlign.

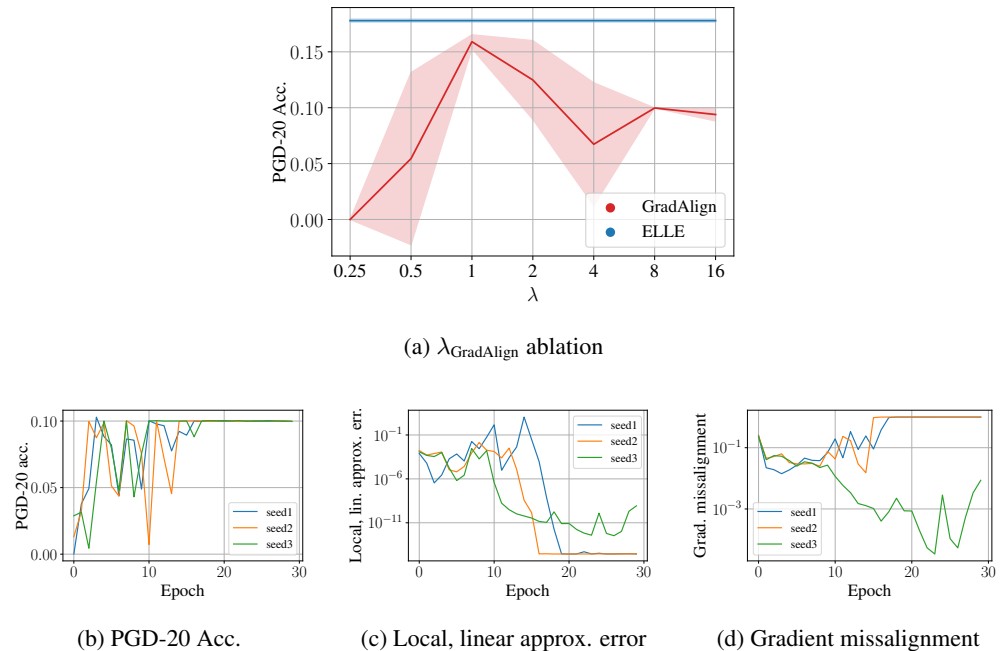

(a) $\lambda_{\text{GradAlign}}$ ablation

(b) PGD-20 Acc.    (c) Local, linear approx. error    (d) Gradient missalignment

Figure 16: **GradAlign ablations: (a):** PGD-20 accuracy for different $\lambda_{\text{GradAlign}}$ values. Only $\lambda_{\text{GradAlign}} = 1$ avoids CO and convergence to a constant classifier. **(b)-(d):** evolution of PGD-20 accuracy, local, linear approximzation error and gradient missalignment for 3 runs with $\lambda_{\text{GradAlign}} = 8$. Gradient missalignment is an unstable metric for close-to-constant classifiers, spiking to 1 when the local, linear approximation error is close to 0.

## B.9 EVALUATION WITH STRONGER ATTACKS

In order to accurately asess the robustness of a model, robustness verification methods should be employed (Ehlers, 2017). However, state-of-the-art robustness verification methods are computationally expensive and donnot scale well to large $\epsilon$ or large models (Zhang et al., 2022a). Alternatively, a common practice is evaluating the resistance to strong adversarial attacks such as PGD-50-10 or the attack ensemble AutoAttack (AA) (Croce and Hein, 2020a).

In this section we confirm our findings from Sec. 4.4, specifically from Fig. 3, by evaluating the AA accuracy of those models. In Table 2 we observe that the methods involving our regularization term, i.e., `ELLE`, `ELLE-A` and `N-FGSM+ELLE-A` are stable and donnot suffer from CO in any setup, while GradAlign and N-FGSM suffer from CO, specially in CIFAR100. Remarkably, adding our regularization term to N-FGSM, helps N-FGSM avoid CO and improves its performance by $+5.17\%$ in SVHN, $+1.07\%$ in CIFAR10 and $+3.20\%$ in CIFAR100 for the largest $\epsilon$.

## B.10 REGULARIZING WITHOUT ATTACKS

In this ablation study, motivated by Moosavi-Dezfooli et al. (2019), we test the performance of our methods when not performing adversarial attacks during training, only enforcing local linearity. We evaluate the performance of PRN when trained with the *Long* schedule on SVHN with standard training, `ELLE` and `ELLE` without attacks. We report the PGD-20 accuracy at $\epsilon \in \{2/255, 8/255, 12/255\}$ over a single run with each method.

In Fig. 17 we observe standard training obtains $0\%$ adversarial accuracy for $\epsilon > 2/255$. Interestingly, when simply plugging our regularizer `ELLE`, we can attain non-trivial robustness. Nevertheless, performance is notably far from the `ELLE` baseline, e.g., $5.07\%$ v.s. $32.17\%$ at $\epsilon = 12/255$. This result is aligned with the findings of Moosavi-Dezfooli et al. (2019): enforcing local linearity uniformly is not enough and the adversarial direction information, e.g., $\nabla_{\boldsymbol{x}}\mathcal{L}(\boldsymbol{f}_{\boldsymbol{\theta}}(\boldsymbol{x}, y))$ needs to be incorporated during training.

Table 2: AutoAttack (AA) accuracy for PRN in SVHN, CIFAR10 and CIFAR100 for the models in Fig. 3.

| SVHN | | | | | | |
|---|---|---|---|---|---|---|
| $\epsilon$ | ELLE | ELLE-A | GradAlign | N-FGSM | N-FGSM+ELLE-A | AT PGD-10 |
| 2 | 88.05 ± (0.12) | 87.66 ± (0.16) | 86.39 ± (0.10) | 88.12 ± (0.16) | **88.24** ± (0.16) | 88.44 ± (0.14) |
| 4 | 73.69 ± (0.16) | 72.13 ± (0.22) | 71.50 ± (0.01) | 74.27 ± (0.27) | **74.75** ± (0.21) | 76.71 ± (0.27) |
| 6 | 57.70 ± (0.27) | 56.19 ± (0.15) | 56.34 ± (0.25) | 59.91 ± (0.23) | **60.05** ± (0.08) | 64.47 ± (0.48) |
| 8 | 41.77 ± (0.23) | 42.55 ± (0.18) | 27.92 ± (19.63) | 46.45 ± (0.20) | **46.70** ± (0.14) | 53.00 ± (0.31) |
| 10 | 30.55 ± (0.26) | 30.96 ± (0.07) | 30.21 ± (0.50) | 33.28 ± (0.30) | **35.33** ± (0.28) | 42.60 ± (0.09) |
| 12 | 22.20 ± (0.08) | 19.97 ± (0.53) | 21.20 ± (0.26) | 21.18 ± (0.33) | **26.35** ± (0.23) | 32.98 ± (0.58) |
| CIFAR10 | | | | | | |
| $\epsilon$ | ELLE | ELLE-A | GradAlign | N-FGSM | N-FGSM+ELLE-A | AT PGD-10 |
| 2 | 78.77 ± (0.16) | 78.71 ± (0.04) | 78.94 ± (0.12) | **79.05** ± (0.08) | 78.64 ± (0.24) | 78.85 ± (0.16) |
| 4 | 65.70 ± (0.12) | 65.64 ± (0.17) | 65.89 ± (0.25) | **66.00** ± (0.06) | 65.83 ± (0.06) | 66.55 ± (0.22) |
| 6 | 53.80 ± (0.18) | 53.97 ± (0.12) | 54.20 ± (0.05) | **54.26** ± (0.18) | 54.17 ± (0.30) | 55.62 ± (0.37) |
| 8 | 42.78 ± (0.95) | 44.32 ± (0.04) | 44.66 ± (0.21) | 44.81 ± (0.18) | **45.05** ± (0.26) | 46.95 ± (0.11) |
| 10 | 33.79 ± (0.41) | 35.55 ± (0.03) | 36.10 ± (0.22) | 37.00 ± (0.20) | **37.16** ± (0.16) | 39.54 ± (0.35) |
| 12 | 27.13 ± (1.30) | 28.17 ± (0.12) | 28.63 ± (0.34) | 30.56 ± (0.12) | **30.61** ± (0.34) | 33.55 ± (0.50) |
| 14 | 21.75 ± (0.49) | 22.19 ± (0.46) | 22.53 ± (0.82) | **25.22** ± (0.41) | 24.86 ± (0.15) | 28.62 ± (0.45) |
| 16 | 18.28 ± (0.17) | 18.03 ± (0.15) | 17.46 ± (1.71) | **20.59** ± (0.21) | 20.48 ± (0.57) | 24.77 ± (0.26) |
| 18 | 14.55 ± (0.26) | 14.75 ± (0.44) | 15.12 ± (0.40) | **16.63** ± (0.44) | 16.36 ± (0.27) | 21.30 ± (0.12) |
| 20 | 13.01 ± (0.09) | **14.80** ± (0.21) | 11.57 ± (1.57) | 13.30 ± (0.33) | 12.88 ± (0.62) | 13.55 ± (7.11) |
| 22 | 13.82 ± (0.17) | 13.58 ± (0.07) | 9.96 ± (0.04) | 11.39 ± (0.48) | **14.15** ± (0.31) | 16.26 ± (0.73) |
| 24 | 12.85 ± (0.28) | 12.99 ± (0.30) | 11.32 ± (1.33) | 11.09 ± (0.82) | **13.15** ± (0.27) | 15.44 ± (0.00) |
| 26 | 11.71 ± (0.41) | **12.25** ± (0.34) | 10.00 ± (0.00) | 10.96 ± (0.26) | 12.03 ± (1.02) | 14.42 ± (0.00) |
| CIFAR100 | | | | | | |
| $\epsilon$ | ELLE | ELLE-A | GradAlign | N-FGSM | N-FGSM+ELLE-A | AT PGD-10 |
| 2 | 47.82 ± (0.37) | 48.01 ± (0.10) | 48.40 ± (0.18) | **48.53** ± (0.33) | 47.99 ± (0.25) | 47.83 ± (0.50) |
| 4 | 34.79 ± (0.19) | 35.68 ± (0.17) | 35.85 ± (0.20) | **36.07** ± (0.36) | 35.90 ± (0.12) | 35.87 ± (0.45) |
| 6 | 26.43 ± (0.27) | 27.44 ± (0.35) | 27.61 ± (0.29) | **27.67** ± (0.36) | 27.54 ± (0.31) | 27.84 ± (0.62) |
| 8 | 20.01 ± (0.11) | 21.04 ± (0.28) | 13.67 ± (9.69) | **21.45** ± (0.24) | 21.26 ± (0.10) | 22.05 ± (0.60) |
| 10 | 15.88 ± (0.21) | 16.39 ± (0.04) | 0.00 ± (0.00) | 17.20 ± (0.13) | **17.24** ± (0.10) | 17.72 ± (0.64) |
| 12 | 13.01 ± (0.15) | 13.29 ± (0.26) | 13.23 ± (0.40) | **14.06** ± (0.17) | 13.96 ± (0.21) | 14.70 ± (0.41) |
| 14 | 10.67 ± (0.02) | 10.88 ± (0.15) | 11.11 ± (0.30) | **11.78** ± (0.16) | 11.55 ± (0.15) | 12.62 ± (0.23) |
| 16 | 8.94 ± (0.12) | 8.93 ± (0.23) | 3.72 ± (3.84) | **9.98** ± (0.16) | 9.78 ± (0.15) | 10.86 ± (0.10) |
| 18 | 7.63 ± (0.07) | 7.33 ± (0.10) | 1.00 ± (0.00) | 8.30 ± (0.02) | **8.51** ± (0.14) | 9.49 ± (0.17) |
| 20 | 6.44 ± (0.01) | 6.21 ± (0.16) | 1.00 ± (0.00) | 4.52 ± (3.20) | **6.99** ± (0.05) | 8.39 ± (0.14) |
| 22 | 5.27 ± (0.32) | 5.26 ± (0.20) | 2.41 ± (2.00) | 3.69 ± (2.65) | **7.54** ± (0.05) | 7.50 ± (0.02) |
| 24 | 4.37 ± (0.06) | 4.39 ± (0.28) | 0.93 ± (0.10) | 1.55 ± (2.18) | **5.10** ± (0.21) | 6.68 ± (0.11) |
| 26 | 3.64 ± (0.14) | 2.87 ± (0.25) | 1.00 ± (0.00) | 1.17 ± (1.65) | **4.37** ± (0.14) | 6.09 ± (0.04) |

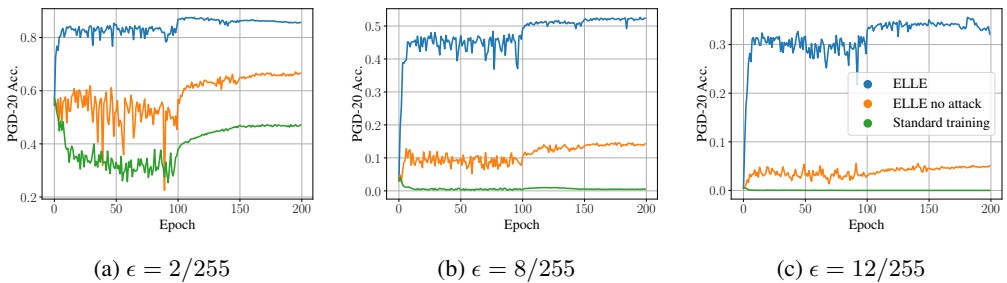

(a) $\epsilon = 2/255$     (b) $\epsilon = 8/255$     (c) $\epsilon = 12/255$

Figure 17: **No attack ablation:** We report the SVHN PGD-20 accuracy for ELLE, ELLE regularization without attacks and standard training at **(a)** $\epsilon = 2/255$, **(b)** $\epsilon = 8/255$ and **(c)** $\epsilon = 12/255$. The legend is shared across plots. Simply enforcing local linearity with our regularization method is not sufficient to attain the optimal adversarial accuracy.

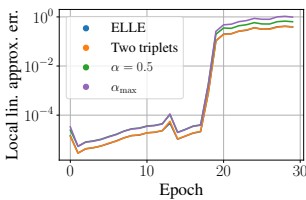
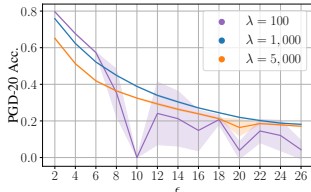

(a) Mean of local linearity metric    (b) ELLE-2p variant on CIFAR10

Figure 18: ELLE **variants:**. **(a)** Mean value of the modifications of our local linearity metric during FGSM training at $\epsilon = 8/255$. All of the variants differ by a constant factor, favoring the the use of the cheapest approaches. **(b)** ELLE-2p variant results. This variant with 1 less point than ELLE in the forward pass is also able to overcome CO. However, no guarantees of enforcing local linearity are present.

### B.11    ELLE-2P: REGULARIZING WITH LESS MEMORY

In this section we reuse $x_{\text{FGSM}}$ instead of the random sample $x_b$ in Algorithm 1. This allows using less memory by only forwarding 3 points instead of 4 per point in the batch. Instead of randomly sampling 2 points $x_a$ and $x_b$ in $\mathbb{B}[x, \epsilon, \infty]$, we sample $x_a \sim x + \text{Unif}\left([-\epsilon, \epsilon]^d\right)$ and $\alpha \sim \text{Unif}([0, 1])$ and compute $x_c = (1 - \alpha) \cdot x_a + \alpha \cdot x_{\text{FGSM}}$. Finally, our new regularization term becomes:

$$\hat{E}_{\text{lin}} = |\mathcal{L}\left(f_{\theta}(x_c), y\right) - (1 - \alpha) \cdot \mathcal{L}\left(f_{\theta}(x_a), y\right) - \alpha \cdot \mathcal{L}\left(f_{\theta}(x_{\text{FGSM}}), y\right)| . \tag{6}$$

We test the performance of this variant when training PRN architectures in CIFAR10 for $\epsilon$ up to $26/255$ (as in Sec. 4.4). We notice lower values of $\lambda$ were needed for this variant and test the performance with $\lambda \in \{100;\ 1,000;\ 5,000\}$, we report the PGD-20 adversarial accuracy over 3 runs.

In Fig. 18b we observe ELLE-2p overcomes CO for $\lambda \in \{1,000;\ 5,000\}$. Nevertheless, the formulation in Eq. (6) does not strictly enforce local linearity, i.e., there are other class of functions that also satisfy $\hat{E}_{\text{lin}} = 0$. We provide an example of a function with zero error which is non-linear in the following.

**Example 1** (Non locally linear function with $\hat{E}_{\text{lin}} = 0$)**.** *We consider the following continuous function $f \in C^0(\mathbb{R}^2)$:*

$$f(x) = \begin{cases} \langle w_1, x \rangle & \text{if } \langle v, x \rangle \geq 0 \\ \langle w_2, x \rangle & \text{if } \langle v, x \rangle < 0 \end{cases}, \tag{7}$$

*where $w_1 = \begin{pmatrix} -1 \\ -1 \end{pmatrix}$, $w_2 = \begin{pmatrix} 0 \\ -3/2 \end{pmatrix}$ and $v = \begin{pmatrix} 2 \\ -1 \end{pmatrix}$. We consider the set $S = [0, 1] \times [0, 1]$ and the point $x = \begin{pmatrix} 1/2 \\ 1/2 \end{pmatrix}$. Then, $\nabla_x f(x) = w_1 = \begin{pmatrix} -1 \\ -1 \end{pmatrix}$, therefore, we have $x_{\text{FGSM}} = x + 1/2 \cdot \text{sign}\left(\nabla_x f(x)\right) = \begin{pmatrix} 0 \\ 0 \end{pmatrix}$. knowing this, it is easy to check that Eq. (6) is zero. For both cases $\langle v, x \rangle \geq 0$ and $\langle v, x \rangle < 0$ our function is linear and Eq. (6) is zero in both regions and then it is zero in the whole set $S$.*

### B.12    ADDITIONAL LLR EXPERIMENTS

As proven in Propositions 1 and 3, both the LLR and ELLE regularization terms approximate second order directional derivatives. Leaving aside the computational advantages of ELLE(-A) displayed in Fig. 1 and Appx. B.7, we would like to analyze the difference in robustness between LLR and ELLE(-A). We run LLR with $\lambda \in \{200,\ 500,\ 1,000,\ 2,000,\ 5,000\}$ and report the validation PGD-20 accuracy.

In Fig. 19 we observe a similar behavior to ELLE in Fig. 7b. For small values of $\lambda$ and big $\epsilon$, CO still appears. When increasing $\lambda$, CO is avoided but performance is degraded. We select the best performing $\lambda$ for each $\epsilon$ value for the LLR evaluation in Fig. 3.

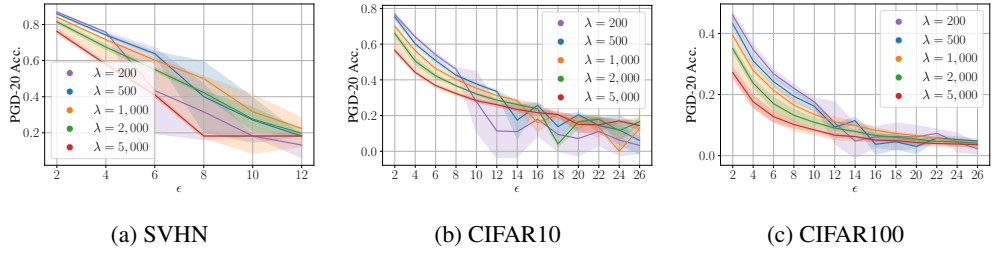

| (a) SVHN | (b) CIFAR10 | (c) CIFAR100 |

Figure 19: Validation PGD-20 accuracy with LLR in SVHN and CIFAR10/100 for different $\lambda$ values.

Table 3: Combination of `ELLE-A` with single-step variants of TRADES and AdvMixUp in CIFAR10. Both AdvMixUp and TRADES suffer from CO for $\epsilon > 2/255$. CO is avoided when `ELLE-A` is plugged in.

| | 2 | | 8 | | 16 | | 26 | |
|---|---|---|---|---|---|---|---|---|
| Method | AA | Clean | AA | Clean | AA | Clean | AA | Clean |
| AdvMixUp-single | **76.06** ± (0.28) | 91.35 ± (0.13) | 0.00 ± (0.00) | 88.61 ± (0.24) | 0.00 ± (0.00) | 86.19 ± (0.09) | 0.00 ± (0.00) | 84.81 ± (0.15) |
| AdvMixUp-single+ELLE-A | 75.97 ± (0.27) | 91.25 ± (0.20) | **38.20** ± (0.48) | 81.83 ± (0.34) | **12.33** ± (0.51) | 56.64 ± (6.81) | **10.80** ± (0.44) | 28.95 ± (0.79) |
| TRADES-single | 63.76 ± (0.09) | 88.50 ± (0.07) | 0.01 ± (0.01) | 90.57 ± (0.01) | 0.00 ± (0.00) | 91.34 ± (0.27) | 0.00 ± (0.00) | 90.92 ± (0.21) |
| TRADES-single+ELLE-A | 62.84 ± (0.43) | 86.21 ± (0.21) | 17.55 ± (1.98) | 73.09 ± (1.37) | 3.16 ± (0.51) | 62.62 ± (2.81) | 1.55 ± (0.52) | 48.26 ± (3.01) |

## B.13 COMBINING ELLE-A WITH ADDITIONAL METHODS

To further showcase the benefits of combining our regularization term, we analyze the performance of single-step variants of AdvMixUp (Lee et al., 2020) and TRADES (Zhang et al., 2019) when in combination with `ELLE-A`. We train PRN in CIFAR10 at $\epsilon \in \{2, 8, 16, 26\}/255$. We use the standard hyperparameters for AdvMixUp and TRADES, for `ELLE-A` we use the standard $\gamma = 0.99$ and $\lambda = 5,000$ for AdvMixUp and $\lambda = 10,000$ for TRADES.

In Table 3, we can observe both the single-step variants of AdvMixUp and TRADES suffer from CO for $\epsilon > 2/255$. Alternatively, CO is avoided when `ELLE-A` is plugged into the training. In Fig. 20, the performance drop during training the single-step variants of AdvMixUp and TRADES is displayed at $\epsilon = 8/255$. When `ELLE-A` is plugged into the training, the PGD-20 accuracy steadily increases during training.

## B.14 ALTERNATIVE SOLUTIONS: APPROXIMATING GRADALIGN WITH FINITE DIFERENCES

A naive solution to Double Backpropagation is approximating the gradient terms in Eq. (5) with Finite Differences approximations (LeVeque, 2007). Let $e_i$ be the $i^{th}$ vector of the canonical basis

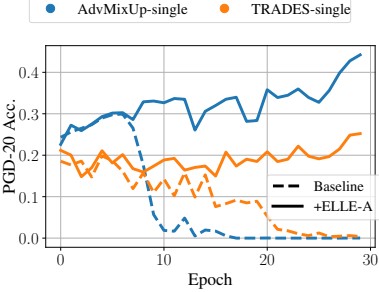

Figure 20: Training evolution with the single-step variants of TRADES and AdvMixUp in CIFAR10 at $\epsilon = 8/255$. Only when in combination with `ELLE-A`, both methods can obtain a final non-zero adversarial accuracy.

and $\sigma \in \mathbb{R}^+$. We can approximate the gradient of the loss function $\mathcal{L}\left(\boldsymbol{f_\theta}(\cdot), y\right) \in C^1(\mathbb{R}^d)$ with the classical finite differences approximation:

$$[\nabla_{\boldsymbol{x}} \mathcal{L}\left(\boldsymbol{f_\theta}(\boldsymbol{x}), y\right)]_i \approx \frac{\mathcal{L}\left(\boldsymbol{f_\theta}(\boldsymbol{x} + \sigma \boldsymbol{e}_i), y\right) - \mathcal{L}\left(\boldsymbol{f_\theta}(\boldsymbol{x}), y\right)}{\sigma} .$$

Sadly, this approximation involves $d+1$ function evaluations. In total, we would need $2(d+1)$ new evaluations, e.g., $6,146$ for SVHN/CIFAR10/100 or $497,666$ for ImageNet, just to estimate the regularization term in a single image in a single iteration of the training process. Nesterov and Spokoiny (2017) study gradient approximations via a single sample of a multivariate Gaussian distribution in the convex optimization setup. This efficient approximation is given by:

$$\boldsymbol{g}(\boldsymbol{x}, \boldsymbol{u}, \sigma) = \frac{\mathcal{L}\left(\boldsymbol{f_\theta}(\boldsymbol{x} + \sigma \boldsymbol{u}), y\right) - \mathcal{L}\left(\boldsymbol{f_\theta}(\boldsymbol{x}), y\right)}{\sigma} \boldsymbol{u} \approx \nabla_{\boldsymbol{x}} \mathcal{L}\left(\boldsymbol{f_\theta}(\boldsymbol{x} + \sigma \boldsymbol{e}_i)\right), \quad \boldsymbol{u} \sim \mathcal{N}(\boldsymbol{0}, \boldsymbol{I}_d).$$

Plugging this approximation into Eq. (5) we get:

$$\text{Eq. (5)} \approx 1 - \text{sign}\left[\left(\mathcal{L}\left(\boldsymbol{f_\theta}(\boldsymbol{x} + \sigma \boldsymbol{u}), y\right) - \mathcal{L}\left(\boldsymbol{f_\theta}(\boldsymbol{x}), y\right)\right)\left(\mathcal{L}\left(\boldsymbol{f_\theta}(\boldsymbol{x} + \boldsymbol{\eta} + \sigma \boldsymbol{v}), y\right) - \mathcal{L}\left(\boldsymbol{f_\theta}(\boldsymbol{x} + \boldsymbol{\eta}), y\right)\right)\right] \frac{\boldsymbol{u}^\top \boldsymbol{v}}{||\boldsymbol{u}|| ||\boldsymbol{v}||},$$
$$(8)$$

where $\boldsymbol{u}, \boldsymbol{v} \sim \mathcal{N}(\boldsymbol{0}, \boldsymbol{I}_d), \boldsymbol{\eta} \sim \text{Unif}([-\epsilon, \epsilon]^d)$.

We notice this approximation is non-smooth due to the $\text{sign}(\cdot)$ operator. This results in the gradient of this term w.r.t. $\boldsymbol{\theta}$ being zero almost everywhere. Therefore is not straightforward to employ Eq. (8) as a regularization term. Moreover, this approximation would need $4$ extra function evaluations, $1$ more than our proposed method.

## C  PROOFS AND ADDITIONAL LOCAL LINEARITY METRICS

In this section we include our proof of Proposition 1. Additionally, we elaborate on how different local, linear approximation error definitions could be proposed with more points and how this reflects on their approximation of the second order derivative. Lastly, a similar proof is included for relating the local linearity definition in Qin et al. (2019) is available in Proposition 3.

*Proof of Proposition 1.* This relationship can be obtained by means of the Taylor series expansion of $h_i$ around $\boldsymbol{x}_c = (1 - \alpha) \cdot \boldsymbol{x}_a + \alpha \cdot \boldsymbol{x}_b$.

Firstly, we note that $\boldsymbol{x}_a$ and $\boldsymbol{x}_b$ can be expressed as:

$$\boldsymbol{x}_a = \boldsymbol{x}_c + \alpha \cdot (\boldsymbol{x}_a - \boldsymbol{x}_b), \quad \boldsymbol{x}_b = \boldsymbol{x}_c - (1 - \alpha) \cdot (\boldsymbol{x}_a - \boldsymbol{x}_b).$$

Now, for $i \in [o]$, by means of the Taylor expansion of $h_i$ at $\boldsymbol{x}_c$ we have:

$$
\begin{aligned}
h_i(\boldsymbol{x}_a) = {} & h_i(\boldsymbol{x}_c) + \alpha \cdot (\boldsymbol{x}_a - \boldsymbol{x}_b)^\top \nabla_{\boldsymbol{x}} h_i(\boldsymbol{x}_c) \\
& + \frac{\alpha^2}{2} \cdot (\boldsymbol{x}_a - \boldsymbol{x}_b)^\top \nabla_{\boldsymbol{xx}}^2 h_i(\boldsymbol{x}_c)(\boldsymbol{x}_a - \boldsymbol{x}_b) \\
& + \frac{\alpha^3}{6} \cdot \sum_{j=1}^d \sum_{k=1}^d \sum_{l=1}^d (x_a - x_b)_j (x_a - x_b)_k (x_a - x_b)_l \frac{\partial^3 h_i(\boldsymbol{x}_c)}{\partial x_j x_k x_l} \\
& + O\left( \|\boldsymbol{x}_a - \boldsymbol{x}_b\|_\infty^4 \right) \\
h_i(\boldsymbol{x}_b) = {} & h_i(\boldsymbol{x}_c) - (1 - \alpha) \cdot (\boldsymbol{x}_a - \boldsymbol{x}_b)^\top \nabla_{\boldsymbol{x}} h_i(\boldsymbol{x}_c) \\
& + \frac{(1 - \alpha)^2}{2} \cdot (\boldsymbol{x}_a - \boldsymbol{x}_b)^\top \nabla_{\boldsymbol{xx}}^2 h_i(\boldsymbol{x}_c)(\boldsymbol{x}_a - \boldsymbol{x}_b) \\
& - \frac{(1 - \alpha)^3}{6} \cdot \sum_{j=1}^d \sum_{k=1}^d \sum_{l=1}^d (x_a - x_b)_j (x_a - x_b)_k (x_a - x_b)_l \frac{\partial^3 h_i(\boldsymbol{x}_c)}{\partial x_j x_k x_l} \\
& + O\left( \|\boldsymbol{x}_a - \boldsymbol{x}_b\|_\infty^4 \right).
\end{aligned} \tag{9}
$$

By operating the inner term in Definition 1, substituting Eq. (9) when applicable and noticing $(\boldsymbol{x}_a - \boldsymbol{x}_b)^\top \nabla_{\boldsymbol{xx}}^2 h_i(\boldsymbol{x}_c)(\boldsymbol{x}_a - \boldsymbol{x}_b) = D_{(\boldsymbol{x}_a - \boldsymbol{x}_b)}^2 h_i(\boldsymbol{x}_c)$ we obtain:

$$
\begin{aligned}
& h_i(\boldsymbol{x}_c) \\
& -(1 - \alpha) \cdot h_i(\boldsymbol{x}_a) \\
& \quad -\alpha \cdot h_i(\boldsymbol{x}_b) = \frac{-\alpha(1 - \alpha)}{2} D_{(\boldsymbol{x}_a - \boldsymbol{x}_b)}^2 h_i(\boldsymbol{x}_c) \\
& \qquad \frac{\alpha(1 - \alpha)(1 - 2\alpha)}{6} \cdot \sum_{j=1}^d \sum_{k=1}^d \sum_{l=1}^d (x_a - x_b)_j (x_a - x_b)_k (x_a - x_b)_l \frac{\partial^3 h_i(\boldsymbol{x}_c)}{\partial x_j x_k x_l} \\
& \qquad + O\left( \|\boldsymbol{x}_a - \boldsymbol{x}_b\|_\infty^4 \right) \\
& \qquad = \frac{-\alpha(1 - \alpha)}{2} D_{(\boldsymbol{x}_a - \boldsymbol{x}_b)}^2 h_i(\boldsymbol{x}_c) + O\left( \|\boldsymbol{x}_a - \boldsymbol{x}_b\|_\infty^3 \right),
\end{aligned} \tag{10}
$$

where in the last equality we used that $(x_a - x_b)_j \le \|\boldsymbol{x}_a - \boldsymbol{x}_b\|_\infty \ \forall j \in [d]$ and that because $h_i \in C^3(\mathbb{R}^d)$ we have $\sum_{j=1}^d \sum_{k=1}^d \sum_{l=1}^d \frac{\partial^3 h_i(\boldsymbol{x}_c)}{\partial x_j x_k x_l} < \infty$. Lastly, by substituting Eq. (10) into Definition 1 the proof is concluded. □

Different local, linear approximation error definitions arise easily when noticing Definition 1 is simply a Finite Differences (FD) approximation (LeVeque, 2007) of the second derivative of $g(\alpha)_i = h_i(\boldsymbol{x}_a + \alpha \cdot (\boldsymbol{x}_b - \boldsymbol{x}_a))$. Therefore, as shown in Chapter 1.5 of (LeVeque, 2007) we can obtain a high order approximation of the $n^{\text{th}}$ derivative provided any set of non-overlapping points $x_1, x_2, \cdots, x_m$ with $m \ge n + 1$. Here, we provide an example with 5 equispaced points.

**Definition 3** (5-point local, linear approximation error). *Let $\boldsymbol{h} :\in \mathbb{R}^d \to \mathbb{R}^o$, let the points:*

$$\boldsymbol{x}_c = \frac{3\boldsymbol{x}_a + \boldsymbol{x}_b}{4}, \quad \boldsymbol{x}_d = \frac{\boldsymbol{x}_a + \boldsymbol{x}_b}{2}, \quad \boldsymbol{x}_e = \frac{\boldsymbol{x}_a + 3\boldsymbol{x}_b}{4}$$

*the 5-point local, linear approximation error of $\boldsymbol{h}$ at $\mathbb{B}[\boldsymbol{x}, \epsilon, p]$ for $1 \le p \le \infty$ is given by:*

$$E_{Lin}(\boldsymbol{h}, \boldsymbol{x}, p, \epsilon) = \underset{\substack{\boldsymbol{x}_a, \boldsymbol{x}_b \sim Unif(\mathbb{B}[\boldsymbol{x}, \epsilon, p]) \\ \alpha \sim Unif([0,1])}}{\mathbb{E}} \left[ \left\| -\frac{1}{12}\boldsymbol{h}(\boldsymbol{x}_a) + \frac{4}{3}\boldsymbol{h}(\boldsymbol{x}_c)) - \frac{5}{2}\boldsymbol{h}(\boldsymbol{x}_d) + \frac{4}{3}\boldsymbol{h}(\boldsymbol{x}_e) - \frac{1}{12}\boldsymbol{h}(\boldsymbol{x}_b) \right\|_2 \right] .$$

$$(11)$$

In the fashion of Proposition 1, we have the following relationsip.

**Proposition 2.** *Let $\boldsymbol{h} \in C^6(\mathbb{R}^d)$ be a six times differentiable mapping, let $E_{Lin}(\boldsymbol{h}, \boldsymbol{x}, p, \epsilon)$ and $D^2_{\boldsymbol{v}}(h_i(\boldsymbol{x}))$ be defined as in Definitions 1 and 2 and $\left[ D^2_{\boldsymbol{v}}(h_i(\boldsymbol{x})) \right]_{i=1}^o \in \mathbb{R}^o$ be the vector containing the second order directional derivatives along direction $\boldsymbol{v}$ for every output coordinate of $\boldsymbol{h}$, the following relationship follows:*

$$E_{Lin}(\boldsymbol{h}, \boldsymbol{x}, p, \epsilon) = \underset{\substack{\boldsymbol{x}_a, \boldsymbol{x}_b \sim Unif(\mathbb{B}[\boldsymbol{x}, \epsilon, p]) \\ \alpha \sim Unif([0,1])}}{\mathbb{E}} \left( \left\| \left[ \frac{1}{25}D^2_{\boldsymbol{x}_a - \boldsymbol{x}_b}(h_i(\boldsymbol{x}_d)) + O(\|\boldsymbol{x}_a - \boldsymbol{x}_b\|^6_\infty) \right]_{i=1}^o \right\|_2 \right) ,$$

$$(12)$$

*where $\boldsymbol{x}_d := \frac{1}{2} \cdot \boldsymbol{x}_a + \frac{1}{2} \cdot \boldsymbol{x}_b$.*

*Proof of Proposition 2.* This relationship can be obtained by means of the Taylor series expansion of $h_i$ around $\boldsymbol{x}_d$. In this proof, we will use the shorthand $\sum_{j,\cdots,m,\cdots,p=1}^d = \sum_{j=1}^d \cdots \sum_{m=1}^d \cdots \sum_{p=1}^d$

Firstly, we notice that we can express any point as a linear combination of $\boldsymbol{x}_d$ and $\boldsymbol{x}_b - \boldsymbol{x}_a$:

$$\boldsymbol{x}_a = \boldsymbol{x}_d + \frac{2}{5} \cdot (\boldsymbol{x}_a - \boldsymbol{x}_b)$$

$$\boldsymbol{x}_b = \boldsymbol{x}_d - \frac{2}{5} \cdot (\boldsymbol{x}_a - \boldsymbol{x}_b)$$

$$\boldsymbol{x}_c = \boldsymbol{x}_d + \frac{1}{5} \cdot (\boldsymbol{x}_a - \boldsymbol{x}_b)$$

$$\boldsymbol{x}_e = \boldsymbol{x}_d - \frac{1}{5} \cdot (\boldsymbol{x}_a - \boldsymbol{x}_b).$$

Now, For $i \in [o]$ , by means of the Taylor expansion of $h_i$ at $\boldsymbol{x}_d$ we have:

$$
\begin{aligned}
h_i(\boldsymbol{x}_a) = h_i(\boldsymbol{x}_d) &+ \frac{2}{5} \cdot (\boldsymbol{x}_a - \boldsymbol{x}_b)^\top \nabla_{\boldsymbol{x}} h_i(\boldsymbol{x}_d) \\
&+ \frac{4}{50} \cdot (\boldsymbol{x}_a - \boldsymbol{x}_b)^\top \nabla^2_{\boldsymbol{x}\boldsymbol{x}} h_i(\boldsymbol{x}_d)(\boldsymbol{x}_a - \boldsymbol{x}_b) \\
&+ \frac{8}{750} \cdot \sum_{j,k,l=1}^d (x_a - x_b)_j (x_a - x_b)_k (x_a - x_b)_l \frac{\partial^3 h_i(\boldsymbol{x}_d)}{\partial x_j x_k x_l} \\
&+ \frac{16}{15000} \cdot \sum_{j,k,l,m=1}^d (x_a - x_b)_j (x_a - x_b)_k (x_a - x_b)_l (x_a - x_b)_m \frac{\partial^4 h_i(\boldsymbol{x}_d)}{\partial x_j x_k x_l x_m} \\
&+ \frac{32}{375000} \cdot \sum_{j,k,l,m,n=1}^d (x_a - x_b)_j (x_a - x_b)_k (x_a - x_b)_l (x_a - x_b)_m (x_a - x_b)_n \frac{\partial^5 h_i(\boldsymbol{x}_d)}{\partial x_j x_k x_l x_m x_n} \\
&+ \frac{64}{11250000} \cdot \sum_{j,k,l,m,n,p=1}^d (x_a - x_b)_j (x_a - x_b)_k (x_a - x_b)_l (x_a - x_b)_m (x_a - x_b)_n (x_a - x_b)_p \frac{\partial^6 h_i(\boldsymbol{x}_d)}{\partial x_j x_k x_l x_m x_n x_p} \\
&+ O\left( \|\boldsymbol{x}_a - \boldsymbol{x}_b\|^7_\infty \right) ,
\end{aligned}
$$

$$(13)$$

$$
\begin{aligned}
h_i(\boldsymbol{x}_b) = h_i(\boldsymbol{x}_d) &- \frac{2}{5} \cdot (\boldsymbol{x}_a - \boldsymbol{x}_b)^\top \nabla_{\boldsymbol{x}} h_i(\boldsymbol{x}_d) \\
&+ \frac{4}{50} \cdot (\boldsymbol{x}_a - \boldsymbol{x}_b)^\top \nabla_{\boldsymbol{x}\boldsymbol{x}}^2 h_i(\boldsymbol{x}_d)(\boldsymbol{x}_a - \boldsymbol{x}_b) \\
&- \frac{8}{750} \cdot \sum_{j,k,l=1}^{d} (x_a - x_b)_j (x_a - x_b)_k (x_a - x_b)_l \frac{\partial^3 h_i(\boldsymbol{x}_d)}{\partial x_j x_k x_l} \\
&+ \frac{16}{15000} \cdot \sum_{j,k,l,m=1}^{d} (x_a - x_b)_j (x_a - x_b)_k (x_a - x_b)_l (x_a - x_b)_m \frac{\partial^4 h_i(\boldsymbol{x}_d)}{\partial x_j x_k x_l x_m} \\
&- \frac{32}{375000} \cdot \sum_{j,k,l,m,n=1}^{d} (x_a - x_b)_j (x_a - x_b)_k (x_a - x_b)_l (x_a - x_b)_m (x_a - x_b)_n \frac{\partial^5 h_i(\boldsymbol{x}_d)}{\partial x_j x_k x_l x_m x_n} \\
&+ \frac{64}{11250000} \cdot \sum_{j,k,l,m,n,p=1}^{d} (x_a - x_b)_j (x_a - x_b)_k (x_a - x_b)_l (x_a - x_b)_m (x_a - x_b)_n (x_a - x_b)_p \frac{\partial^6 h_i(\boldsymbol{x}_d)}{\partial x_j x_k x_l x_m x_n x_p} \\
&+ O\left( \|\boldsymbol{x}_a - \boldsymbol{x}_b\|_\infty^7 \right),
\end{aligned}
\tag{14}
$$

$$
\begin{aligned}
h_i(\boldsymbol{x}_c) = h_i(\boldsymbol{x}_d) &+ \frac{1}{5} \cdot (\boldsymbol{x}_a - \boldsymbol{x}_b)^\top \nabla_{\boldsymbol{x}} h_i(\boldsymbol{x}_d) \\
&+ \frac{1}{50} \cdot (\boldsymbol{x}_a - \boldsymbol{x}_b)^\top \nabla_{\boldsymbol{x}\boldsymbol{x}}^2 h_i(\boldsymbol{x}_d)(\boldsymbol{x}_a - \boldsymbol{x}_b) \\
&+ \frac{1}{750} \cdot \sum_{j,k,l=1}^{d} (x_a - x_b)_j (x_a - x_b)_k (x_a - x_b)_l \frac{\partial^3 h_i(\boldsymbol{x}_d)}{\partial x_j x_k x_l} \\
&+ \frac{1}{15000} \cdot \sum_{j,k,l,m=1}^{d} (x_a - x_b)_j (x_a - x_b)_k (x_a - x_b)_l (x_a - x_b)_m \frac{\partial^4 h_i(\boldsymbol{x}_d)}{\partial x_j x_k x_l x_m} \\
&+ \frac{1}{375000} \cdot \sum_{j,k,l,m,n=1}^{d} (x_a - x_b)_j (x_a - x_b)_k (x_a - x_b)_l (x_a - x_b)_m (x_a - x_b)_n \frac{\partial^5 h_i(\boldsymbol{x}_d)}{\partial x_j x_k x_l x_m x_n} \\
&+ \frac{1}{11250000} \cdot \sum_{j,k,l,m,n,p=1}^{d} (x_a - x_b)_j (x_a - x_b)_k (x_a - x_b)_l (x_a - x_b)_m (x_a - x_b)_n (x_a - x_b)_p \frac{\partial^6 h_i(\boldsymbol{x}_d)}{\partial x_j x_k x_l x_m x_n x_p} \\
&+ O\left( \|\boldsymbol{x}_a - \boldsymbol{x}_b\|_\infty^7 \right),
\end{aligned}
\tag{15}
$$

$$
\begin{aligned}
h_i(\boldsymbol{x}_e) = h_i(\boldsymbol{x}_d) &- \frac{1}{5} \cdot (\boldsymbol{x}_a - \boldsymbol{x}_b)^\top \nabla_{\boldsymbol{x}} h_i(\boldsymbol{x}_d) \\
&+ \frac{1}{50} \cdot (\boldsymbol{x}_a - \boldsymbol{x}_b)^\top \nabla_{\boldsymbol{x}\boldsymbol{x}}^2 h_i(\boldsymbol{x}_d)(\boldsymbol{x}_a - \boldsymbol{x}_b) \\
&- \frac{1}{750} \cdot \sum_{j,k,l=1}^{d} (x_a - x_b)_j (x_a - x_b)_k (x_a - x_b)_l \frac{\partial^3 h_i(\boldsymbol{x}_d)}{\partial x_j x_k x_l} \\
&+ \frac{1}{15000} \cdot \sum_{j,k,l,m=1}^{d} (x_a - x_b)_j (x_a - x_b)_k (x_a - x_b)_l (x_a - x_b)_m \frac{\partial^4 h_i(\boldsymbol{x}_d)}{\partial x_j x_k x_l x_m} \\
&- \frac{1}{375000} \cdot \sum_{j,k,l,m,n=1}^{d} (x_a - x_b)_j (x_a - x_b)_k (x_a - x_b)_l (x_a - x_b)_m (x_a - x_b)_n \frac{\partial^5 h_i(\boldsymbol{x}_d)}{\partial x_j x_k x_l x_m x_n} \\
&+ \frac{1}{11250000} \cdot \sum_{j,k,l,m,n,p=1}^{d} (x_a - x_b)_j (x_a - x_b)_k (x_a - x_b)_l (x_a - x_b)_m (x_a - x_b)_n (x_a - x_b)_p \frac{\partial^6 h_i(\boldsymbol{x}_d)}{\partial x_j x_k x_l x_m x_n x_p} \\
&+ O\left( \|\boldsymbol{x}_a - \boldsymbol{x}_b\|_\infty^7 \right).
\end{aligned}
\tag{16}
$$

By operating the inner term in Definition 3, substituting Eqs. (13) to (16) when applicable and noticing $(\boldsymbol{x}_a - \boldsymbol{x}_b)^\top \nabla^2_{\boldsymbol{x}\boldsymbol{x}} h_i(\boldsymbol{x}_d)(\boldsymbol{x}_b - \boldsymbol{x}_a) = D^2_{(\boldsymbol{x}_b - \boldsymbol{x}_a)} h_i(\boldsymbol{x}_d)$ we obtain:

$$
\begin{aligned}
-&\frac{1}{12}\boldsymbol{h}(\boldsymbol{x}_a) + \frac{4}{3}\boldsymbol{h}(\boldsymbol{x}_c)) - \frac{5}{2}\boldsymbol{h}(\boldsymbol{x}_d) + \frac{4}{3}\boldsymbol{h}(\boldsymbol{x}_e) - \frac{1}{12}\boldsymbol{h}(\boldsymbol{x}_b) \\
=&\frac{1}{25} D^2_{(\boldsymbol{x}_b - \boldsymbol{x}_a)} h_i(\boldsymbol{x}_d) \\
&- \frac{8}{11250000} \cdot \sum_{j,k,l,m,n,p=1}^{d} (x_a - x_b)_j (x_a - x_b)_k (x_a - x_b)_l (x_a - x_b)_m (x_a - x_b)_n (x_a - x_b)_p \frac{\partial^6 h_i(\boldsymbol{x}_d)}{\partial x_j x_k x_l x_m x_n x_p} \\
&+ O\left(\|\boldsymbol{x}_a - \boldsymbol{x}_b\|_\infty^7\right) \\
=&\frac{1}{25} D^2_{(\boldsymbol{x}_a - \boldsymbol{x}_b)} h_i(\boldsymbol{x}_c) + O\left(\|\boldsymbol{x}_a - \boldsymbol{x}_b\|_\infty^6\right),
\end{aligned}
$$
(17)

where in the last equality we used that $(x_a - x_b)_j \le \|\boldsymbol{x}_a - \boldsymbol{x}_b\|_\infty \; \forall j \in [d]$ and that because $h_i \in C^6(\mathbb{R}^d)$ we have $\sum_{j,k,l,m,n,p=1}^{d} \frac{\partial^6 h_i(\boldsymbol{x}_c)}{\partial x_j x_k x_l x_m x_n x_p} < \infty$. Lastly, by substituting Eq. (10) into Definition 1 the proof is concluded. $\qquad\square$

Notice that in the proofs of Propositions 1 and 2 we require as many degrees of differentiability as the lowest order of accuracy of the FD formula, i.e, the highest order Taylor series term that cannot be cancelled. Regardless of this requirement for the proof, both Definitions 1 and 3 can be applied to any mapping $h : \mathbb{R}^d \to \mathbb{R}^o$. Lastly, we prove a relationship between the regularization term in Qin et al. (2019).

**Proposition 3.** *Let $h \in C^3(\mathbb{R}^d)$ be a tree times differentiable mapping, let $g(\boldsymbol{x}_a, \boldsymbol{x}_b) = h(\boldsymbol{x}_a) - h(\boldsymbol{x}_b) - (\boldsymbol{x}_a - \boldsymbol{x}_b)^\top \nabla_{\boldsymbol{x}} h(\boldsymbol{x}_b)$ and $D^2_{\boldsymbol{v}}(h_i(\boldsymbol{x}))$ be defined as in Definition 2, the following relationship follows:*

$$
g(\boldsymbol{x}_a, \boldsymbol{x}_b) = \frac{1}{2} D^2_{(\boldsymbol{x}_a - \boldsymbol{x}_b)} h_i(\boldsymbol{x}_b) + O\left(\|\boldsymbol{x}_a - \boldsymbol{x}_b\|_\infty^3\right).
$$
(18)

*Proof of Proposition 3.* Similarly to the proof of Propositions 1 and 2, this relationship can be obtained by means of the Taylor series expansion of $h$ around $\boldsymbol{x}_b$. We have:

$$
\begin{aligned}
h(\boldsymbol{x}_a) =& h(\boldsymbol{x}_b) + (\boldsymbol{x}_a - \boldsymbol{x}_b)^\top \nabla_{\boldsymbol{x}} h(\boldsymbol{x}_b) + \frac{1}{2}(\boldsymbol{x}_a - \boldsymbol{x}_b)^\top \nabla^2_{\boldsymbol{x}\boldsymbol{x}} h(\boldsymbol{x}_b)(\boldsymbol{x}_a - \boldsymbol{x}_b) \\
&+ \frac{1}{6} \cdot \sum_{j=1}^{d}\sum_{k=1}^{d}\sum_{l=1}^{d} (x_a - x_b)_j (x_a - x_b)_k (x_a - x_b)_l \frac{\partial^3 h(\boldsymbol{x}_b)}{\partial x_j x_k x_l} \\
&+ O\left(\|\boldsymbol{x}_a - \boldsymbol{x}_b\|_\infty^4\right).
\end{aligned}
$$
(19)

Plugging Eq. (19) into $g$ we have:

$$
g(\boldsymbol{x}_a, \boldsymbol{x}_b) = \frac{1}{2}(\boldsymbol{x}_a - \boldsymbol{x}_b)^\top \nabla^2_{\boldsymbol{x}\boldsymbol{x}} h(\boldsymbol{x}_b)(\boldsymbol{x}_a - \boldsymbol{x}_b) + O\left(\|\boldsymbol{x}_a - \boldsymbol{x}_b\|_\infty^3\right),
$$
(20)

and the proof is finalized. $\qquad\square$

