# OpenReview forum: "Efficient local linearity regularization to overcome catastrophic overfitting"
_ICLR.cc/2024/Conference — ICLR 2024 poster_

### Official Review · Reviewer_e1KG · 2023-10-17

**Soundness:** 2 fair
**Presentation:** 2 fair
**Contribution:** 1 poor
**Rating:** 5
**Confidence:** 4

**Summary:**

This paper aims to mitigate the catastrophic overfitting (CO) phenomenon in adversarial training by enforcing local linearity in the underlying model. Their proposed solution (ELLE) is a computationally efficient approach to enforce local linearity without using double backpropagation, which is time-intensive. Instead, this approach enforces local linearity using multiple forward passes, which is memory-intensive, and hence trading off time for space. Experiments show that their proposed approach is faster than the alternatives, and is successful at mitigating CO.

**Strengths:**

- This paper proposes a simple but effective strategy to enforce local linearity that does not incur double backpropagation costs. Their theoretical results also show that their method implicitly reduces a quantity related to the Hessian, which is expected.

- The paper has a nice study on the CO problem in the experiments section, especially where it defines “perturbation-wise” and “duration-wise” CO, and has nice figures demonstrating CO and its avoidance with the provided method.

**Weaknesses:**

**Not conceptually novel, fails to discuss simpler solutions**

The main idea presented is not conceptually novel. Local linearity has been presented as a solution to catastrophic overfitting several times in the literature (which the paper also discusses). The only novelty is regularizing for linearity using an objective that does not involve gradients. I am unsure how novel or interesting this part itself is – for instance, one can just as well use any other local linearity regularizer (for example, LLR or GradAlign), and replace the gradient terms therein using finite differences. This would eliminate the need for the “double backpropagation” as well and meet their desiderata. The paper needs to discuss why these simple solutions are undesirable if at all they are.

**Misses key references**

The paper misses a couple of key references (see [1,2]), which aim to achieve the same objective as this given paper. It would be great if the authors could comment on how their method compares with the approaches presented in these works.

[1] Singla et al., Low Curvature Activations Reduce Overfitting in Adversarial Training, 2021
[2] Srinivas et al., Efficient Training of Low-Curvature Neural Networks, 2022

**Experiments ignore a canonical adversarial defense: PGD**

While the experimental section nicely demonstrates CO and the effect of its method, it **fails to present comparisons** with the canonical method of PGD [Madry et al., 2018]. While speed comparisons are made with PGD in Figure 1, there do not appear to be accuracy comparisons (AA, Clean accuracy) or comparisons with regard to the ability of PGD to locally linearize the model or usage of the proposed method (ELLE) with PGD. This notable omission of PGD significantly weakens the paper. Rather, the paper focuses primarily on improving FGSM-based defenses (although it also discusses improving GAT and N-FGSM in the experiments), which is a significantly weaker defense than PGD

**Questions:**

- The authors claim in Section 3.3 that the GAT and NuAT methods are not relevant since they attempt to make models locally constant as opposed to locally linear. But surely, local constancy is a special case of local linearity, and thus it is not inconceivable to compare the two methods on equal grounds, like you do for LLR and GradAlign?


- The authors are encouraged to comment on the highlighted weaknesses of the paper.

---

> ### Author Response · Authors · 2023-11-15
> **Rebuttal**
>
> We thank the reviewer e1KG for their feedback. We have made the changes in the text visible in red. We address below their concerns.
>
> - **(Q1): Not conceptually novel**
>
> **(A1):** We respectfully disagree with this statement. Admittedly, the spirit of enforcing local linearity has been well applied as a solution to CO. However, there are several key contributions introduced in our paper:
> - Our method is the first to avoid the 3x time cost coming from Double Backpropagation while being based on the same local linearity principle.
> - We theoretically link our regularization term and LLR [1] as approximants of second-order directional derivatives (Prop. 1 and 3). Compared to LLR, ELLE poses the advantage of not requiring double backpropagation.
> - We develop the first method to adaptively regularize local linearity and showcase its benefits.
> - We showcase the appearance of Delayed CO for several other methods and demonstrate that adding our regularization can resolve this issue.
>
> While our approach maintains simplicity, the results in the paper demonstrate the robustness of the method to tackle CO and outperform SOTA for large $\epsilon$. If the reviewer has noticed any of the previous contributions in related papers, we are happy to reevaluate this list.
>
> - **(Q2): Why not use Finite Differences to approximate LLR or GradAlign?**
>
> **(A2):** Let us explain why Finite Differences don’t fit well in this setup. We theoretically link LLR and ELLE by showing both regularization terms approximate second-order directional derivatives (see Prop. 3 and 1). Therefore, using a Finite Differences approximation of the gradient in the LLR case is unnecessary.
>
> We elaborate on the approximation of GradAlign with Finite Differences in Appendix B.14. Estimating the gradients via Finite Differences is computationally expensive with $d+1$ function evaluations needed for estimating every gradient ($d = 248,832$ for ImageNet). Additionally, when more efficient approximations are involved [2], the regularization term becomes non-smooth due to the appearance of the sign function and it is not directly differentiable, as we would desire for a regularization term. Moreover, it requires one more function evaluation than ELLE. If the reviewer has in mind an alternative way to compute the finite differences, we would be happy to discuss this further.
>
> - **(Q3): Missing References.**
>
> **(A3):** We are thankful for the references that link curvature and robust generalization. We accordingly cite [3] and [4] in the introduction. However, we emphasize that there exist distinct differences between these papers and our method.
>
> [3] analyze the role of the curvature of the activation functions in robust generalization when performing AT. Our method is not based on architectural design, but the optimization method used to train a given architecture. The combination of low curvature architecture design and regularization terms to enforce local linearity is an interesting future direction.
>
> [4] propose a method for controlling the global curvature of twice differentiable neural networks to improve robust generalization. Our method controls the local curvature of the network and is not restricted to twice-differentiable classifiers.
>
> - **(Q4): Experiments ignore a canonical adversarial defense: PGD**
>
> **(A4):** AT is an effective but expensive defense. This is precisely the reason that faster single-step alternatives flourished [6,7], trading off performance for a 10x speed up. The main target of our work is handling critical failures and weaknesses of single-step adversarial training methods. Ideally, we would like single-step methods as reliable as AT PGD-10 but without the computational burden of PGD.
>
> To address the reviewer’s concern, we have included the AT PGD-10 evaluation in [Fig 1](https://imgur.com/a/xjV7Bdc), [Fig 2](https://imgur.com/a/sMToIKd), [Fig 3](https://imgur.com/a/aFpRbkz) and [Fig 5](https://imgur.com/a/e7Z8mQ1). The ability of AT PGD-10 to generate locally linear models is well known and displayed in Fig. 2, where both the gradient misalignment and our local linear approximation error are controlled during training. N-FGSM+ELLE-A is the method that attains the closest performance to AT PGD-10.
>
> - **(Q5): Isn’t enforcing local constancy (GAT and NuAT) a special case of local linearity?**
>
> **(A5):** We thank the reviewer for the attentive reading. We have revised the text in section 3.3.
>
> Let us clarify that in the original submission, we already had comparisons against GAT. As studied in [5], GAT and NuAT are not able to overcome CO for large values of $\epsilon$. In [Fig 5](https://imgur.com/a/e7Z8mQ1), we show that GAT suffers from CO for all cases except PRN at $\epsilon=8/255$. When combined with ELLE-A, GAT training is stabilized and CO does not appear.
>
> We will be happy to answer any other comments or questions the reviewer might have.

---

> > ### Author Response · Authors · 2023-11-15
> > **References**
> >
> > [1] Chongli Qin et al., Adversarial robustness through local linearization, NeurIPS 2019
> >
> > [2] Nesterov and Spokoiny, Random Gradient-Free Minimization of Convex Functions, Foundations of Computational Mathematics, 2017
> >
> > [3] Singla et al., Low Curvature Activations Reduce Overfitting in Adversarial Training, ICCV 2021
> >
> > [4] Srinivas et al., Efficient Training of Low-Curvature Neural Networks, NeurIPS 2022
> >
> > [5] de Jorge et al., Make Some Noise: Reliable and Efficient Single-Step Adversarial Training, NeurIPS 2022
> >
> > [6] Shafahi et al., Adversarial Training for Free!, NeurIPS 2019
> >
> > [7] Wong et al., Fast is better than free: Revisiting adversarial training, ICLR 2020

---

> > > ### Author Response · Authors · 2023-11-18
> > > **Have the concerns of the reviewer been addressed?**
> > >
> > > Dear reviewer e1KG,
> > >
> > > We are grateful for the time spent on the review. The reviewer was concerned about a) the possibility of approximating LLR and GradAlign with Finite Differences b) missing references c) absence of multi-step AT in the comparisons d) writing of the related work section.
> > >
> > > Our rebuttal addresses these points by discussing a) why it is not a good idea to do Finite differences approximations of LLR and GradAlign b) highlighting the differences between our work and the references provided c) Including the AT PGD-10 in our comparisons d) correcting the sentence mentioned in the related work section.
> > >
> > > If the concerns of the reviewer are addressed, we would appreciate it if the reviewer increases their score. If the reviewer still has any remaining concerns, we are more than happy to clarify further.
> > >
> > > Best regards,
> > >
> > > Authors

---

> > > > ### Comment · Reviewer_e1KG · 2023-11-21
> > > >
> > > > Dear authors,
> > > >
> > > > Thank you for the rebuttal, and the experiments with PGD.
> > > >
> > > > (A2) " a Finite Differences approximation of the gradient in the LLR case is unnecessary"; "Estimating the gradients via Finite Differences is computationally expensive with function evaluations"
> > > >
> > > > Please note; finite-difference style approaches are necessary primarily to enable efficient computation, in your case, to avoid the double backpropogation problem. These approaches have been used in the literature before, for example (Moosavi-Dezfooli et al., 2019), which the paper also cites in section 3.3.
> > > >
> > > > (A3) "Our method controls the local curvature of the network and is not restricted to twice-differentiable classifiers"
> > > >
> > > > Please note; the curvature of models is only defined for twice differentiable functions. The theory presented in your paper (e.g.: definition 2) also assumes twice differentiability.
> > > >
> > > > ----
> > > >
> > > > The new experiments with PGD slightly strengthen the paper, so I will increase my score, but due to the reasons presented in my original review, I still cannot advocate for its acceptance.

---

> > > > > ### Author Response · Authors · 2023-11-21
> > > > > **Remaining clarifications**
> > > > >
> > > > > We are grateful to the reviewer e1KG’s response, for acknowledging our changes and increasing the score. Regarding the remaining points, we provide the following insights:
> > > > >
> > > > > - **(P1): ”finite-difference style approaches are necessary primarily to enable efficient computation”, “These approaches have been used in the literature before, for example (Moosavi-Dezfooli et al., 2019)”**
> > > > >
> > > > > **(A1)** Our finite differences approximation of the second order directional derivative indeed saves computation, just like in CURE or LLR. Notice that CURE and LLR are also finite differences approximations of curvature-related terms. We **agree with the reviewer** in this sense, in fact, our method further exploits this fact by not requiring gradient terms.
> > > > >
> > > > > However, notice that in our rebuttal we showed that approximating the gradient terms from GradAlign and LLR with Finite Differences, goes against this intuition and is not practically implementable.
> > > > >
> > > > > - **(P2): “The theory presented in your paper (e.g.: definition 2) also assumes twice differentiability”**
> > > > >
> > > > > **(A2)** We agree with the reviewer on the theoretical part on twice differentiability. Our regularization term has the nice property that when assuming twice differentiability, second order directional derivatives are approximated. The same assumption is made in [1] for CURE. Regardless of the theoretical analysis of CURE and our method relying on twice-differentiability, both CURE and our method can be successfully applied to non-twice differentiable classifiers like ReLU NNs as we show in the experiments.
> > > > >
> > > > > The method proposed in [2] involves the maximum value of the second derivative of the activation function for the computations. They upper bound it with $\beta$ in their formulation. This is not defined for ReLU NNs as they are not twice-differentiable and cannot be applied.
> > > > >
> > > > > We again thank the reviewer for their time. We remain available for further discussion.
> > > > >
> > > > > Regards,
> > > > > Authors
> > > > >
> > > > > **References:**
> > > > >
> > > > > [1] Moosavi-Dezfooli, Robustness via curvature regularization, and vice versa, CVPR 2019
> > > > >
> > > > > [2] Srinivas et al., Efficient Training of Low-Curvature Neural Networks, NeurIPS 2022

---

> > > > > > ### Author Response · Authors · 2023-11-23
> > > > > > **Which are the remaining concerns of the reviewer?**
> > > > > >
> > > > > > Dear reviewer e1KG,
> > > > > >
> > > > > > We thank the reviewer for initially responding to our rebuttal and increasing the score. However, we respectfully believe the current grading is not justified as all of the concerns of the reviewer have been addressed. We kindly ask you to consider proposing for acceptance.
> > > > > >
> > > > > > Regards,
> > > > > >
> > > > > > Authors

---

### Official Review · Reviewer_vQnz · 2023-10-28

**Soundness:** 3 good
**Presentation:** 3 good
**Contribution:** 3 good
**Rating:** 6
**Confidence:** 3

**Summary:**

The paper proposes a new regularization techinque which address the catasterophic overfitting of AT approaches.
The new requalrization term is combined with FGSM AT method to improve the robustness of the model during training.

**Strengths:**

The method addresses a significant challenge in the field of adversarial training with a fairly simple yet effective approach.
I appreciate the theoretical analysis provided in the paper and the level of details shared by the authors.

**Weaknesses:**

I understand the paper mainly focuses on the theoretical aspect of the proposed algorithm.
However, there are some aspects missing which make it difficult to understand how practical the proposed method is.

1- There are several SOTA AT methods currently being used which have demonstrated significant improvement in this area. It would be benficial to have other SOTA adversarial attacks evaluation in the paper as well to show how this approach helps against other SOTA  attacks as well.

**Questions:**

The main question is that how the proposed method is compared with SOTA adversarial attack algorithms and how this method can be combined with other AT algorithm than FGSM and whether it would be effective in the sence or not.

---

> ### Author Response · Authors · 2023-11-15
> **Rebuttal**
>
> We thank the reviewer vQnz for their feedback. We have made the changes in the text visible in red. We address below their concerns.
>
> - **(Q1): Can you include the evaluation with other SOTA adversarial attacks?**
>
> **(A1):** Our models are evaluated with AutoAttack [1] (See Tab. 1 and 2 and Fig. 5), which contains the SOTA attacks FAB [2] and Square [3]. Moreover, AutoAttack is the standard evaluation metric in the [RobustBench](https://robustbench.github.io/) benchmark. We are not aware of more effective adversarial attacks than the AutoAttack ensemble. We will be happy to include any other SOTA adversarial attacks in the evaluation if the reviewer has any concrete methods in mind.
>
> - **(Q2): Can you integrate your regularization term with other SOTA adversarial training methods other than FGSM?**
>
> **(A2):** Yes, please let us explain how we have already integrated ELLE to other methods. In this paper, we present a thorough evaluation against SOTA single-step adversarial training methods. ELLE(-A) can be integrated as a plug-in regularization term into any other adversarial training method. This is demonstrated in the paper by combining ELLE-A with FGSM, GAT [4], and N-FGSM [5] (See Fig. 3 to 5). As we demonstrate, our method is effective to overcome CO when combined with all these methods.
>
> Additionally, during the rebuttal we have included an analysis of the integration of ELLE-A with single-step variants of AdvMixUp [6] and TRADES [7]. These two methods are only able to overcome CO when in combination with ELLE-A, see [Tab. 3](https://imgur.com/a/lhXoSZN) and [Fig. 20](https://imgur.com/a/SkItddC).
>
> We will be happy to answer further comments the reviewer might have.
>
> **References:**
>
> [1] Croce and Hein, Reliable evaluation of adversarial robustness with an ensemble of diverse parameter-free attacks, ICML 2020
>
> [2] Croce and Hein. Minimally distorted adversarial examples with a fast adaptive boundary attack, ICML 2020
>
> [3] Andriushchenko et al., Square Attack: a query-efficient black-box adversarial attack via random search, ECCV 2020
>
> [4] Sriramanan et al., Guided Adversarial Attack for Evaluating and Enhancing Adversarial Defenses, NeurIPS 2020
>
> [5] de Jorge et al., Make Some Noise: Reliable and Efficient Single-Step Adversarial Training, NeurIPS 2022

---

> > ### Comment · Reviewer_vQnz · 2023-11-22
> >
> > I thank the authors for the additional clarification and the newly added results.

---

> ### Author Response · Authors · 2023-11-18
> **Have the concerns of the reviewer been addressed?**
>
> Dear reviewer vQnz,
>
> We are thankful for the time invested in reviewing our work. The reviewer was concerned about a) the attacks used for evaluating our models b) the possibility of combining our regularization term with other SOTA robust training methods. Our rebuttal further motivates the use of AutoAttack, clarifies the usage of our method in combination with GAT and N-FGSM and extended to combine with AdvMixUp and TRADES.
>
> If the concerns of the reviewer are addressed, we would appreciate it if the reviewer increases their score. If the reviewer still has any remaining concerns, we are more than happy to clarify further.
>
> Best regards,
>
> Authors

---

> ### Author Response · Authors · 2023-11-22
> **Thankful to the Reviewer vQnz**
>
> Dear Reviewer vQnz,
>
> we are grateful for your feedback and appreciation of our work. We are thankful for increasing the score.
>
> Best,
>
> Authors

---

### Official Review · Reviewer_4XDh · 2023-10-30

**Soundness:** 3 good
**Presentation:** 3 good
**Contribution:** 3 good
**Rating:** 6
**Confidence:** 5

**Summary:**

In this paper, the authors investigate the issue of catastrophic overfitting and propose an efficient and effective solution in the form of a local linear regularizer. The introduced algorithm not only mitigates catastrophic overfitting but also circumvents the double backpropagation problem. It performs well in challenging scenarios, such as dealing with large adversarial perturbations and long training schedules, especially when pursuing Fast-AT. Experimental results demonstrate that the proposed method achieves good results while effectively addressing catastrophic overfitting.

**Strengths:**

1. The proposed regularization scheme as well as the analysis are reasonable, and the introduced local linear approximation scheme is very simple.
2. The experiments show that the proposed ELLE can work well under large adversarial perturbations and long training schedules.

**Weaknesses:**

1. The definition of Equation 1 is excessively simplistic, lacking any explanation or validation. It's important to note that the majority of deep models and training losses do not conform to linear mappings, which leads me to question this particular configuration. Also, the authors claim that ''CO appears when the loss suddenly becomes non-linear, ...'' (on page 4) I find this description somewhat confusing. Since the loss function usually remains constant throughout the training process, how does this transition from linear to non-linear occur in the loss?

1. The proposed local linear approximation error is similar to the mixup, which has been studied in adversrial training [a]. But, the comparison and analysis are missing.

1. According to the results shown in Figure 4, I found that ELLE-A is not better than ELLE, but I cannot find any clear explanation.

1. The proposed method does not work well under short training schedule and small adversarial perturbation. The authors have not provided a satisfactory explanation or analysis for this issue. Since the defensive performance obtained from long training schedule and short training schedule is comparable, why should we necessarily opt for the longer training schedule? Moreover, large perturbations imply that the noise becomes more pronounced, which contradicts real-world scenarios.

1. In the review process, I attempted to use mathematical tools to show that the regularization introduced in this paper can be used for the approximate estimation of LLR. That is,

   $L(f_\theta(x+\delta),y)-L(f_\theta(x),y)-\delta^T\nabla_xL(f_\theta(x),y)\approx L(f_\theta(x_c),y)-(1-\alpha)L(f_\theta(x_a),y)-\alpha L(f_\theta(x_b),y)$.

   So, I think ELLE is an effective approximation for LLR. And it's better to compare LLR and ELLE in the all experiments. BTW, the authors only report some results in Figure 1, when considering methods enforcing local linearity. But, I cannot find any comparison in the experiments. I think this lack of comparison to be unfair.

[a] Adversarial Vertex Mixup: Toward Better Adversarially Robust Generalization. CVPR 2020.

**Questions:**

Please check the problems mentioned in the Weaknesses part.

---

> ### Author Response · Authors · 2023-11-15
> **Rebuttal**
>
> We thank the reviewer 4XDh for their feedback. We have made the changes in the text visible in red. We address below their concerns.
>
> - **(Q1): Equation (1) is overly excessively simplistic, additional explanations are needed. Deep models and training losses do not conform to linear mappings.**
>
> **(A1):** We thank the reviewer for highlighting this issue. We agree with the reviewer in that deep models and training losses are usually non-linear. However, even though they are globally non-linear, it has been observed that the loss of robust models behaves linearly in small regions around data points, see [1] and related citations in the third paragraph of the introduction. As an example, the output of Deep NNs with the ReLU activation function is highly non-linear, but piecewise linear, and thus they are locally linear in small regions.
>
> We have clarified this in the preamble of equation (1) and accordingly updated equation (1) to better capture this intuition.
>
> - **(Q2): The loss function usually remains constant throughout the training process, how does this transition from linear to non-linear occur in the loss?**
>
> **(A2):**  “The loss function usually remains constant”: We believe the reviewer refers to the minimization objective, e.g., the cross entropy loss. This is indeed kept the same throughout the training process in AT [2]. Nevertheless, the behavior of the loss landscape with respect to the input data evolves in time depending on the parameters of the model [1]. Let $h(\theta_t, x, y) = L(f_{\theta_t}(x), y)$, we are interested on the local linearity of $h$ w.r.t. $x$. The loss transitions from locally linear to non locally linear in the sense that for some $t = t_1$, we have that $h$ behaves linearly with respect to $x$, but it does not for some other $t = t_2 > t_1$. This behavior is clearly displayed in Fig. 2 (b-c), where the local linear approximation error suddenly increases at epoch 15 when training with FGSM.
>
> - **(Q3): The local, linear approximation error resembles the AdvMixUp formulation, why do you not compare against it?**
>
> **(A3):** We are thankful to the reviewer for the suggestion. We conduct the related experiment in Appendix B.13. The convex combination used in the design of the input points and labels given by AdvMixUp resembles the convex combination used for obtaining the point $x_c$ in our method. However, there are fundamental differences between AdvMixup and ELLE. Let $\delta_{\text{adv}}$ be the adversarial perturbation given by some adversarial attack:
>
> - AdvMixUp assigns the one-hot vector corresponding to the true class to  $x$ and uniform class probabilities to the point $x + 2\delta_{\text{adv}}$. Then, every point in the convex combination of $x$ and $x + 2\delta_{\text{adv}}$ is assigned a convex combination of the aforementioned labels.
> - ELLE enforces the training loss to be locally linear in the whole region $x + \delta: ||\delta|| \leq \epsilon$ via a loss involving a convex combination of terms.
> - In the single-step scenario, i.e. $\delta_{\text{adv}} = \epsilon \cdot \text{sign}(\nabla_{x}L(f_{\theta}(x), y))$, AdvMixUp suffers from CO while our method does not.
>
> We include an experimental evaluation of AdvMixUp and its combination with ELLE-A in Appendix B.13. Showing that combined with ELLE-A, CO is avoided, see [Tab. 3](https://imgur.com/a/lhXoSZN) and [Fig. 20](https://imgur.com/a/SkItddC).
>
> - **(Q4): ELLE-A is worse than ELLE in the long schedule but it is not present in the text.**
>
> **(A4):** We thank the reviewer for highlighting this missing explanation in the text. This is expected because the ELLE-A regularization is weaker. So overall, the model can still overfit in long schedules. ELLE is a stronger regularization and test performance decreases much less with time. We have included a related sentence in section 4.4.
>
> - **(Q5): The proposed regularization method does not perform well in small epsilon values and short schedules.**
>
> **(A5):** Guarding against attacks with a small perturbation budget doesn’t guarantee the resistance against attacks with a larger budget. Additionally, large perturbation radiuses can still produce imperceptible perturbations as we show in [Fig. 6](https://imgur.com/a/QamLafl) for ImageNet. Small radiuses are also not so relevant when studying CO because it does not appear. In fact, even standard FGSM with $\epsilon=2/255$ can be used for ImageNet training without the appearance of CO, see Tab. 1.
>
> The biggest improvements with our method occur when $\epsilon$ is large. However, when combining ELLE-A with N-FGSM, AA accuracy is even improved for $\epsilon \leq 8/255$ in SVHN and is comparable for CIFAR10/100, see Table 2 in the appendix.

---

> ### Author Response · Authors · 2023-11-15
> **Rebuttal (continuation)**
>
> - **(Q6): Why use long schedules and large $\epsilon$?**
>
> **(A6):** Long schedules have been used for other existing robust training methods [4,5]. Other works argue the avoidance of CO in short schedules gives a false sense of robustness [6,7]. Additionally, the evaluation in long schedules is a standard practice [1,3]. The avoidance of CO in longer schedules is an additional benefit of our method. When training in less studied datasets where the appropriate length of the schedule is unknown, our method is safer against CO.
>
> Regarding larger $\epsilon$, we should aim at being robust to the largest possible perturbations. Our attack visualization (See [Fig. 6](https://imgur.com/a/QamLafl)) does not reveal a change in the human prediction with the $\epsilon$ values employed.
>
> - **(Q7): ELLE appears to be an approximation of LLR, LLR should be included in the main comparisons.**
>
> **(A7):** We thank the reviewer for the suggestion. Both LLR and ELLE approximate the second order derivatives up to constant factors. This is proven in Prop. 3 (End of the appendix) and Prop. 1 respectively. Considering this fact, the lower computational cost of ELLE favors its usage. The comparison against LLR is usually not included in the literature [1,3]. Moreover, our results in Fig. 1 show that LLR attains a significantly lower adversarial accuracy than ELLE.
>
> We are currently running additional experiments on CIFAR100 and SVHN. We will post those results and revise the respective tables during the rebuttal period. We include an ablation on the $\lambda$ parameter for LLR in Appendix B.12, [Fig. 19](https://imgur.com/a/SXLapTO), and the corresponding CIFAR10 results in [Fig. 3 (b)](https://imgur.com/a/Gz20Tfn). LLR does not match the performance of ELLE(-A), especially for large $\epsilon$.
>
> We will be happy to answer any other comments or questions the reviewer might have.
>
> **References:**
> [1] Andriushchenko and Flammarion, Understanding and improving fast adversarial training, NeurIPS 2020
>
> [2] Madry et al., Towards Deep Learning Models Resistant to Adversarial Attacks, ICLR 2018
>
> [3] de Jorge et al., Make Some Noise: Reliable and Efficient Single-Step Adversarial Training, NeurIPS 2022
>
> [4] Rice et al., Overfitting in adversarially robust deep learning, ICML 2020.
>
> [5] Xu et al., Exploring and exploiting decision boundary dynamics for adversarial robustness, ICLR 2023
>
> [6] Kim et al., Understanding catastrophic overfitting in single-step adversarial training, AAAI 2021
>
> [7] Li et al., Towards Understanding Fast Adversarial Training, arXiv 2020

---

> ### Author Response · Authors · 2023-11-18
> **Have the concerns of the reviewer been addressed?**
>
> Dear reviewer 4XDh,
>
> We are thankful for your effort to review our work. The main concerns expressed are the a) non-linear mappings learning, b) comparisons with AdvMixUp, and c) comparisons with LLR and d) the need for long schedules and large $\epsilon$. Our rebuttal addresses those core concerns along with the rest of the questions of the reviewer.
>
> In addition, we present have concluded the LLR experiments promised in our first response and include them in ([Fig. 19](https://imgur.com/a/Z1hWOTB)) and its evaluation ([Fig. 3](https://imgur.com/a/uFzb5d0)). We find LLR is more sensitive to hyperparameters and is not able to match the performance of ELLE(-A).
>
> If the concerns of the reviewer are addressed, we would appreciate it if you re-evaluate our submission. If the reviewer still has any remaining concerns, we are more than happy to clarify further.
>
> Best regards,
>
> Authors

---

> > ### Comment · Reviewer_4XDh · 2023-11-23
> > **Comments on the authors' response**
> >
> > Thank the authors for addressing most of my concerns, and they have corrected them in the newly submitted version. Overall, I would like to increase my score.

---

> > > ### Author Response · Authors · 2023-11-23
> > > **Thank you for acknowledging our revisions**
> > >
> > > Dear Reviewer 4XDh,
> > >
> > > thank you for your time and feedback in our submission. Your feedback has enabled us to improve our paper, e.g., by including experiments with AdvMixUp, LLR and clarifying the different techniques. Please let us know if there are any remaining concerns.
> > >
> > > Best,
> > >
> > > Authors

---

### Author Response · Authors · 2023-11-22
**End of discussion window approaching**

Dear reviewers 4XDh and vQnz,

During this rebuttal we have included **additional experiments** with AT PGD-10, LLR, TRADES and AdvMixUp. We have as well **added clarifications in the text** addressing your concerns and even **discussed alternative methods**.

Since the discussion period is coming to an end and reviewer e1KG updated their score, we would like to remind you to update the score if your concerns have as well been addressed.

Regards,

Authors

---

### Meta-Review · Area_Chair_uTse · 2023-12-05

**Metareview:**

This paper is concerned with improving single-step adversarial training approaches to avoid catastrophic overfitting. Building upon the argument that local linearity is a necessary condition for successful adversarial training, authors propose an efficient regularization strategy that bias the learning algorithm towards models satisfying that property, and do so by replacing a number of forward-backward cycles as required in multi step adversarial training by an approach using a single-step attacker followed by a few extra forward passes.

Reviewers highlighted the importance of the topic under consideration, but highlighted concerns with novelty and significance of the empirical evaluation. In particular, it was highlighted that there exist competing approaches to enforcing local linearity, and the proposal does not outperform multi-step pgd. However, as highlighted by the authors, the proposal matches multi-step adversarial training with a significantly reduced overhead, and the evaluation suggests that the proposed gradient-free linearity enforcing regularizer works as intended. We thus conclude those results are relevant to the community and recomend acceptance.

**Justification For Why Not Higher Score:**

While the paper is interesting and relevant, the empirical assessment could have included a detailed side-by-side empirical comparison of different methods targeting local linearity.

**Justification For Why Not Lower Score:**

N/A

---

### Decision · Program_Chairs · 2024-01-16

Accept (poster)